# Elevated circulating follistatin associates with an increased risk of type 2 diabetes

Chuanyan Wu[1,2,3,23], Yan Borné[1,23], Rui Gao[2], Maykel López Rodriguez[4,5], William C. Roell[6], Jonathan M. Wilson [6], Ajit Regmi[6], Cheng Luan[1], Dina Mansour Aly[1], Andreas Peter[7,8,9], Jürgen Machann [8,9,10], Harald Staiger[7,8,9], Andreas Fritsche[7,8,9], Andreas L. Birkenfeld [7,8,9], Rongya Tao[11], Robert Wagner [7,8,9], Mickaël Canouil [12], Mun-Gwan Hong [13,24], Jochen M. Schwenk [13,24], Emma Ahlqvist [1], Minna U. Kaikkonen [5], Peter Nilsson [1], Angela C. Shore [14], Faisel Khan[15], Andrea Natali[16], Olle Melander[1], Marju Orho-Melander[1], Jan Nilsson [1], Hans-Ulrich Häring[7,8,9], Erik Renström[1], Claes B. Wollheim[1,17], Gunnar Engström[1], Jianping Weng[18], Ewan R. Pearson[19,24], Paul W. Franks [1,24], Morris F. White[11], Kevin L. Duffin[6], Allan Arthur Vaag[20], Markku Laakso [4,21], Norbert Stefan [7,8,9], Leif Groop [1,22] & Yang De Marinis [1,2,18✉]

The hepatokine follistatin is elevated in patients with type 2 diabetes (T2D) and promotes hyperglycemia in mice. Here we explore the relationship of plasma follistatin levels with incident T2D and mechanisms involved. Adjusted hazard ratio (HR) per standard deviation (SD) increase in follistatin levels for T2D is 1.24 (CI: 1.04–1.47, $p < 0.05$) during 19-year follow-up ($n = 4060$, Sweden); and 1.31 (CI: 1.09–1.58, $p < 0.01$) during 4-year follow-up ($n = 883$, Finland). High circulating follistatin associates with adipose tissue insulin resistance and non-alcoholic fatty liver disease ($n = 210$, Germany). In human adipocytes, follistatin dose-dependently increases free fatty acid release. In genome-wide association study (GWAS), variation in the glucokinase regulatory protein gene (*GCKR*) associates with plasma follistatin levels ($n = 4239$, Sweden; $n = 885$, UK, Italy and Sweden) and GCKR regulates follistatin secretion in hepatocytes in vitro. Our findings suggest that GCKR regulates follistatin secretion and that elevated circulating follistatin associates with an increased risk of T2D by inducing adipose tissue insulin resistance.

A full list of author affiliations appears at the end of the paper.

F ollistatin is a secreted protein that is expressed in almost all tissues. It is linked to metabolic diseases[1,2], with elevated plasma levels in patients with type 2 diabetes (T2D)[1]. Evidence suggests that follistatin has multiple auto- and paracrine functions in various tissues. Follistatin binds and neutralizes TGF-β family members[3]. Follistatin is essential for the formation and growth of muscle fibers[4,5] and is involved in the development of muscle fiber hypertrophy[6–11]. Furthermore, follistatin was found to enhance thermogenic gene expression in differentiated mouse brown adipocytes[12]. Short-term follistatin treatment reduced glucagon secretion from islets of Langerhans, whereas long-term follistatin treatment prevented apoptosis and induced proliferation of rat β cells[13]. Local overexpression of follistatin in the pancreas from diabetic mice resulted in increased serum insulin levels[14]. In humans, circulating follistatin derives predominately from the liver and its expression and secretion are upregulated by a high glucagon-to-insulin ratio[13].

In mice[15], follistatin was identified as a mediator of diabetes by promoting white adipose tissue insulin resistance. In hyperglycemic and high-fat-fed obese mice, disruption of follistatin restored glucose tolerance, white adipose tissue insulin signaling and suppression of hepatic glucose production by insulin. In obese individuals with diabetes who underwent gastric bypass surgery, serum follistatin decreased in parallel with HbA$_{1c}$ levels[15]. Circulating follistatin levels were also found to be increased in individuals with T2D, associating positively with HbA$_{1c}$ and fasting blood glucose levels. However, follistatin is unaffected by acute alterations in blood glucose and blood insulin concentrations during an oral glucose tolerance test (OGTT)[1]. So far it is unknown whether elevation of plasma follistatin associates with the risk of T2D, independently of established diabetes risk markers. Furthermore, it needs to be established whether and to what extent genetics explains the variability of circulating follistatin levels and whether the mechanisms of follistatin to induce insulin resistance in mice may also be operative in humans.

In this work, we show that circulating follistatin associates with an increased risk of T2D, independently of established risk factors. A possible mechanism may involve an effect of follistatin to induce adipose tissue insulin resistance, resulting in increased adipocyte free fatty acid (FFA) release, which also promotes nonalcoholic fatty liver disease (NAFLD). Secretion of follistatin from the liver cells is regulated by GCKR, in addition to insulin and glucagon.

## Results

**Plasma follistatin levels associate with the risk of T2D.** We evaluated the association of circulating follistatin levels at baseline with incident T2D in the Malmö Diet and Cancer Cardiovascular Cohort (MDC-CC). Of 4195 participants, 577 (13.75%) individuals developed T2D during a mean (±SD) follow-up time of 19.07 (±5.09) years (Table 1, Supplementary Fig. 1, see Supplementary Information for cohort details). Subjects who developed T2D during follow-up had higher plasma follistatin levels at baseline, compared to those who did not progress to diabetes (Table 1). Circulating follistatin associated with an elevated risk of diabetes, hazard ratio (HR) per 1-SD increase in follistatin levels for T2D is 1.29 (CI: 1.19–1.40, $p < 0.001$), adjusted for age and sex; and 1.24 (CI: 1.04–1.47, $p < 0.05$) adjusted for multiple risk factors (Table 2). When subjects were divided into quartiles (Q) based on plasma follistatin level, the HRs adjusted for age and sex for Q4 vs. Q1 for incident diabetes was 1.97 (CI: 1.55–2.50, $p$ for trend < 0.001); and 1.35 (CI: 1.04–1.74, $p$ for trend = 0.02) adjusted for multiple risk factors (Table 2, Supplementary Table 1 and Fig. 1). Plasma follistatin levels correlated with measures of glucose tolerance, insulin secretion, and insulin sensitivity, before and after adjustment for multiple risk factors (Supplementary Table 2). The C-statistics value, which is a measure of the adequacy of fit for the binary outcomes, for model 1 adjusted for age and sex, was 0.5419 (CI: 0.5184–0.5655) and increased significantly to 0.5832 (CI: 0.5603–0.6060) when follistatin was added to the model (difference in C-statistics, 0.041; CI: 0.0187–0.0637; $p < 0.001$). The C-statistics value adjusted for multiple risk factors was 0.7701 (CI: 0.7510–0.7892) and increased to 0.7718 (CI: 0.7527–0.7909), when follistatin was added (difference in C-statistics, 0.0017; $p = 0.113$), which may suggest that follistatin does not improve diabetes prediction on its own beyond established risk factors.

We also analyzed an independent cohort, IMI-DIRECT-METSIM (see Supplementary Information for cohort details). Among 1079 subjects, 53 (4.91%) developed T2D during the follow-up period (4-year). Individuals with diabetes incidence had higher baseline plasma follistatin levels (Table 3). The HRs per SD of plasma follistatin for diabetes was 1.35 (CI: 1.13–1.61,

**Table 1 Baseline characteristics of MDC-CC study population ($n = 4195$).**

| | Whole study population | No incident diabetes | Incident diabetes | $p$ value [a] |
|---|---|---|---|---|
| **Number ($n$)** | 4195 | 3618 | 577 | – |
| **Follistatin (NPX)[†]** | 26.91 ± 1.43 | 26.54 ± 1.42 | 28.84 ± 1.44 | <0.001 |
| **Age (years)** | 57.33 ± 5.97 | 57.35 ± 6.01 | 57.24 ± 5.70 | 0.696 |
| **Sex (men) $n$ (%)** | 1614 (38.5) | 1370 (37.9) | 244 (42.3) | 0.043 |
| **Waist circumference (cm)** | 82.35 ± 12.14 | 81.51 ± 11.70 | 87.63 ± 13.45 | <0.001 |
| **BMI (kg/m²)** | 25.41 ± 3.70 | 25.15 ± 3.49 | 27.05 ± 4.49 | <0.001 |
| **HDL (mmol/L)** | 1.41 ± 0.37 | 1.42 ± 0.37 | 1.30 ± 0.34 | <0.001 |
| **LDL (mmol/L)** | 4.17 ± 0.98 | 4.14 ± 0.97 | 4.34 ± 1.01 | <0.001 |
| **TG (mmol/L)** | 1.27 ± 0.60 | 1.23 ± 0.57 | 1.51 ± 0.70 | <0.001 |
| **CRP (mg/L)*** | 1.30 (0.60-2.60) | 1.20 (0.60-2.50) | 1.70 (0.90-3.50) | <0.001 |
| **Systolic blood pressure (mmHg)** | 140.10 ± 18.60 | 139.31 ± 18.42 | 145.11 ± 18.98 | <0.001 |
| **BP lowering medication $n$ (%)** | 601 (14.3) | 462 (12.8) | 139 (24.1) | <0.001 |
| **Lipid lowering medication $n$ (%)** | 87 (2.1) | 68 (1.9) | 19 (3.3) | 0.027 |
| **Smoking habits $n$ (%)** | | | | 0.343 |
| Never smokers | 1724 (41.1) | 1497 (41.4) | 227 (39.3) | |
| Ex-smokers | 1563 (37.3) | 1344 (37.1) | 219 (38.0) | |
| Current smokers | 908 (21.6) | 777 (21.5) | 131 (22.7) | |

Values expressed are means (±SD) or percentages unless specified elsewise. [a]One-way analysis of variance (continuous variables) and Pearson's Chi2 (dichotomous variables) for incident diabetes (yes/no). *Median [25–75%]. [†]Follistatin is expressed as linear Normalized Protein eXpression (NPX) AU for relative quantification according to Olink guidance.

**Table 2 Incidence of diabetes in relation to sex-specific quartiles of plasma follistatin levels in MDC-CC.**

|  | Q1 | Q2 | Q3 | Q4 | *p* for trend | HR per SD |
|---|---|---|---|---|---|---|
| **Number of participants** | 1048 | 1049 | 1050 | 1048 |  |  |
| **Incidence of diabetes n (n per 1000 p-y)** | 108 (5.16) | 132 (6.54) | 149 (7.52) | 188 (9.87) |  |  |
| **Follistatin (NPX)†** | 17.27 ± 1.21 | 23.92 ± 1.07 | 30.06 ± 1.08 | 42.22 ± 1.23 |  |  |
| *Model 1 HR* | 1 | 1.28 (0.99–1.66) | 1.47** (1.15–1.89) | 1.97*** (1.55–2.50) | <0.001 | 1.29*** (1.19–1.40) |
| ***Model 2 HR*** | 1 | 1.12 (0.86–1.45) | 1.21 (0.94–1.57) | 1.45** (1.13–1.86) | 0.003 | 1.31** (1.11–1.56) |
| ***Model 3 HR*** | 1 | 1.10 (0.85–1.43) | 1.16 (0.89–1.50) | 1.35* (1.04–1.74) | 0.020 | 1.24* (1.04–1.47) |

Model 1: Adjusted for age and sex (*n* = 4195). Model 2: Adjusted for age, sex, BMI, physical activity, alcohol intake, fiber intake, smoking habits, use of antihypertensive medications, systolic blood pressure, LDL, HDL cholesterol, use of lipid lowering medications, fasting glucose (*n* = 4060). Model 3: Model 2 and CRP (*n* = 4060). The relationship between follistatin levels in plasma and incidence of diabetes during the follow up was explored in 4195 individuals, and fully-adjusted association study was performed in 4060 individuals. HR, hazard ratio. *$p < 0.05$, **$p < 0.01$, ***$p < 0.001$. †Follistatin is expressed as linear Normalized Protein eXpression (NPX) AU for relative quantification according to Olink guidance.

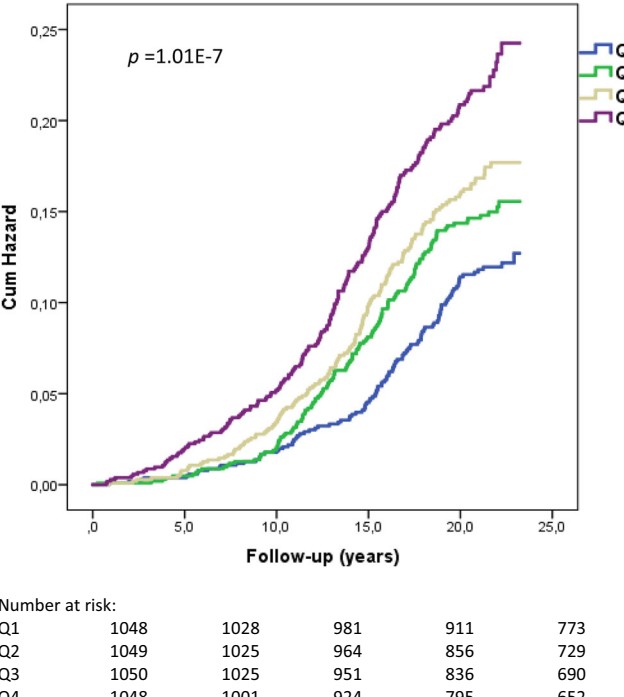

*p* =1.01E-7

Number at risk:
| | | | | | |
|---|---|---|---|---|---|
| Q1 | 1048 | 1028 | 981 | 911 | 773 |
| Q2 | 1049 | 1025 | 964 | 856 | 729 |
| Q3 | 1050 | 1025 | 951 | 836 | 690 |
| Q4 | 1048 | 1001 | 924 | 795 | 652 |

**Fig. 1 Kaplan–Meier curve to illustrate incidence of diabetes in relation to follistatin quartiles.** Time axis was follow-up time until death, emigration, incident diabetes or end of follow-up (*n* = 4195; *p* = 1.01E-7; two-sided log-rank test).

$p < 0.01$) adjusted for age, and 1.31 (CI: 1.09–1.58, $p < 0.01$) adjusted for multiple risk factors (Table 4).

Thus, our findings in longitudinal cohorts indicate that circulating follistatin associates with the risk of developing T2D.

**Relationships of follistatin with adipose tissue insulin resistance and related traits.** In the MDC-CC cohort, we observed an association between the elevated follistatin and insulin resistance (Supplementary Table 2). In animals follistatin was found to contribute to diabetes by promoting adipose tissue insulin resistance and attenuating insulin-inhibited white adipose tissue lipolysis, resulting in increased circulating levels of free fatty acids (FFAs)[15]. Here we investigated relationships of circulating follistatin with adipose tissue insulin resistance and related traits in humans. In subjects of the Tübingen Diabetes Family Study (TDFS) cohort without diabetes (*n* = 210, see Supplementary Information for cohort details), plasma follistatin levels correlated positively with FFAs, measured before and during the OGTT, and

visceral fat mass. Importantly, the relationships of plasma follistatin with fasting (std. ß = 0.17, $p = 0.009$), 60 minutes (std. ß = 0.26, $p < 0.0001$), and 120 minutes (std. ß = 0.27, $p < 0.0001$) FFAs were independent of age, sex, and total body fat mass. Furthermore, plasma follistatin levels correlated negatively with adipose tissue insulin sensitivity and leg fat mass (Supplementary Tables 4, 5 and Fig. 2a). Elevated follistatin levels also associated with higher liver fat content, after adjustment for age, sex, and total body fat mass. Furthermore, circulating follistatin levels, adjusted for age, sex, and total body fat mass, were found to be elevated in patients with nonalcoholic fatty liver disease (NAFLD, Fig. 2b). The relationship of follistatin levels with liver fat content disappeared after further adjustment for visceral fat mass and leg fat mass, or for adipose tissue insulin sensitivity (Supplementary Table 6). Thus, these findings suggest that circulating follistatin independently associates with adipose tissue insulin resistance in humans, and that the relationship of follistatin with fatty liver is possibly explained by effects of follistatin on insulin sensitivity of adipose tissue in the leg and the visceral compartment.

**Effect of follistatin on insulin-mediated suppression of lipolysis in adipocytes.** The effect of follistatin on insulin-mediated suppression of lipolysis was investigated in vitro. Human adipocyte-derived stem cells were differentiated into adipocytes, and treated with 0, 0.3, 3, or 30 μg/mL follistatin for 2 h before exposure to insulin for 3 h. Insulin-inhibited lipolysis was attenuated by follistatin dose-dependently, measured by stepwise increase of glycerol release, reflecting FFA release from adipocytes into the culture medium from the breakdown of stored triglycerides (Fig. 2c).

**GWAS for plasma follistatin levels.** To identify causative factors that influence plasma follistatin levels, we performed GWAS within the MDC-CC cohort (*n* = 4239). We identified 13 single-nucleotide polymorphisms (SNPs), which significantly ($p < 5E-8$) associated with plasma follistatin levels (Fig. 3a and b; Supplementary Table 7). The strongest association was observed for a noncoding SNP rs780094 in the glucokinase regulatory protein (*GCKR*) gene ($p = 1.1E-11$). Two additional SNPs within the *GCKR* gene, rs780093 (noncoding) and rs1260326 (coding), in linkage disequilibrium (LD) with rs780094[16], also showed strong association with plasma follistatin levels, after adjustment for age and sex ($p = 1.91E-11$ and $p = 2.77E-11$; Supplementary Table 7). Other SNPs displaying associations with plasma follistatin, are shown in Supplementary Table 7 and Supplementary Fig. 2. The association between SNPs in the *GCKR* gene and plasma follistatin was replicated in an independent SUMMIT-VIP cohort (*n* = 885, see Supplementary Information for cohort details), where *GCKR* rs1260326 showed the strongest association with

**Table 3 Baseline characteristics of non-diabetic individuals ($n = 1079$) in the IMI-DIRECT-METSIM cohort (Kuopio, Finland).**

|  | No incident diabetes | Incident diabetes | p value[a] |
|---|---|---|---|
| Number | 1026 | 53 |  |
| Follistatin (NPX) | 3991.1 ± 1259.73 | 4329.25 ± 1809.51 | 0.063 |
| Age (years) | 60.97 ± 5.53 | 60.66 ± 5.78 | 0.70 |
| Sex (men) n (%) | 1026 (100) | 53 (100) |  |
| BMI (kg/m$^2$) | 27.63 ± 3.6 | 30.19 ± 5.07 | 8.92E-07 |
| C-peptide (pmol/L) | 812.6 ± 302.24 | 1021.89 ± 428.06 | 1.80E-06 |
| Glucose (mg/dL) | 102.82 ± 7.87 | 113.33 ± 8.75 | 2.53E-20 |
| Glucose 48-month (mg/dL) | 107.28 ± 8.35 | 130.31 ± 3.77 | 8.74E-76 |
| Glucose increase (mg/dL) | 4.45 ± 7.69 | 16.98 ± 8.72 | 6.42E-29 |
| HbA$_{1c}$ (%) | 5.53 ± 0.27 | 5.61 ± 0.31 | 0.033 |
| HDL (mg/dL) | 53.82 ± 14.57 | 49.79 ± 11.52 | 0.048 |
| LDL (mg/dL) | 117.25 ± 33.79 | 119.35 ± 29.35 | 0.66 |
| TG (mg/dL) | 122.06 ± 60.15 | 142.45 ± 54.39 | 0.016 |
| CRP (mg/L) | 1.82 ± 3.58 | 2.26 ± 2.73 | 0.37 |

Values expressed are means (±SD) or percentages unless specified elsewise. [a]One-way analysis of variance (continuous variables) and Pearson's Chi$^2$ (dichotomous variables) for incident diabetes (yes/no). *CRP* C-reactive protein.

**Table 4 Incidence of diabetes in relation to quartiles of follistatin in the IMI-DIRECT-METSIM cohort. The relationship between follistatin levels in plasma and incidence of diabetes during the follow up was explored in 1079 individuals, and fully-adjusted association study was performed in 883 individuals.**

|  | Q1 | Q2 | Q3 | Q4 | p for trend | HR per SD |
|---|---|---|---|---|---|---|
| Number of participants | 270 | 270 | 269 | 270 |  |  |
| Incidence of diabetes n (n per 1000 p-y) | 10 (9.26) | 13 (12.04) | 11 (10.22) | 19 (17.59) |  |  |
| Follistatin (NPX) | 2664.94 ± 343.49 | 3447.24 ± 205.82 | 4194.21 ± 258.59 | 5725.14 ± 1163.54 |  |  |
| Model 1 HR | 1 | 1.69 (0.80–3.58) | 1.49 (0.70–3.22) | 2.61** (1.29–5.30) | 0.011 | 1.35** (1.13–1.61) |
| Model 2 HR | 1 | 1.12 (0.48–2.60) | 1.19 (0.52–2.75) | 2.25* (1.04–4.86) | 0.029 | 1.28** (1.075–1.524) |
| Model 3 HR | 1 | 1.13 (0.49–2.62) | 1.23 (0.53–2.86) | 2.34* (1.07–5.101) | 0.0237 | 1.31** (1.09–1.58) |

Model 1: Adjusted for age ($n = 1079$).
Model 2: Adjusted for age, BMI, physical activity, alcohol intake, fiber intake, LDL, HDL cholesterol, fasting glucose ($n = 883$).
Model 3: Model 2 and CRP ($n = 883$).
*HR* hazard ratio. *$p < 0.05$, **$p < 0.01$.

follistatin levels ($p = 2.5E-10$, Fig. 3c–d; Supplementary Table 8). The variant rs780094 in the *GCKR* gene was found to associate with diabetes in the DIAGRAM GWAS consortium, both in the European (OR, 1.04; 95% CI: 1.00–1.08, $p = 0.025$) and in transethnic cohorts ($p = 1 \times 10^{-5}$)[17].

**Regulation of follistatin secretion in the human hepatocyte cell line HEPG2 BY GCKR.** It has been previously shown that the liver is the major site of follistatin secretion and that follistatin release from the liver is increased by glucagon and inhibited by insulin[13]. Here we studied how GCKR regulates liver follistatin secretion in the human hepatocyte cell line HepG2. GCKR forms a tight complex with GCK in the nucleus, and dissociation of the GCK-GCKR binding leads to increased GCK translocation from the nucleus to the cytoplasm, which regulates liver cell glucose uptake and glycolysis[18–20]. Furthermore, *GCKR* rs1260326 (p.P446L) has a decreased degree of nuclear localization, enabling to sequester GCK and to directly interact with GCK, which elevates hepatic glucose uptake and disposal, by increasing active cytosolic GCK[19]. Here we applied GCKR-GCK transfection in combination with a chemical disruptor AMG-3969, to model the proposed effects for the functional SNPs that we identified in GWAS, i.e., effects on GCKR-GCK binding and GCKR expression.

HepG2 cells were transfected with GCK or co-transfected with GCK and GCKR expressing plasmids (Supplementary Fig. 3).

Forty-eight hours after transfection, the cells were treated with and without AMG-3669, a GCK-GCKR complex disruptor that promotes translocation of disassociated GCK from the nucleus to the cytoplasm[21]. The addition of AMG-3969 induced GCK translocation and produced a similar localization pattern to that of free GCK. The nuclear localization of GCK was highest in the absence of the disruptor (Supplementary Figs. 4, 5). In the presence of the GCK-GCKR complex and the complex disruptor AMG-3969, glucagon increased follistatin secretion by 40%, compared to control (Fig. 4a). This increase was inhibited by insulin (Fig. 4b). Transfection with GCK alone, or GCK-GCKR co-transfection without AMG-3969, had no effect on follistatin secretion. Thus, our results support the notion that follistatin secretion is regulated by GCKR in addition to insulin and glucagon.

## Discussion

In this study, we observed that plasma follistatin levels were elevated many years prior to the onset of T2D, and that circulating follistatin at baseline associated with incident T2D, independently of established diabetes risk markers. We also found that follistatin associated with adipose tissue insulin resistance and related traits, and that follistatin attenuated insulin-mediated suppression of lipolysis in adipocytes. Furthermore, we performed GWAS and identified variants in the *GCKR* to be the

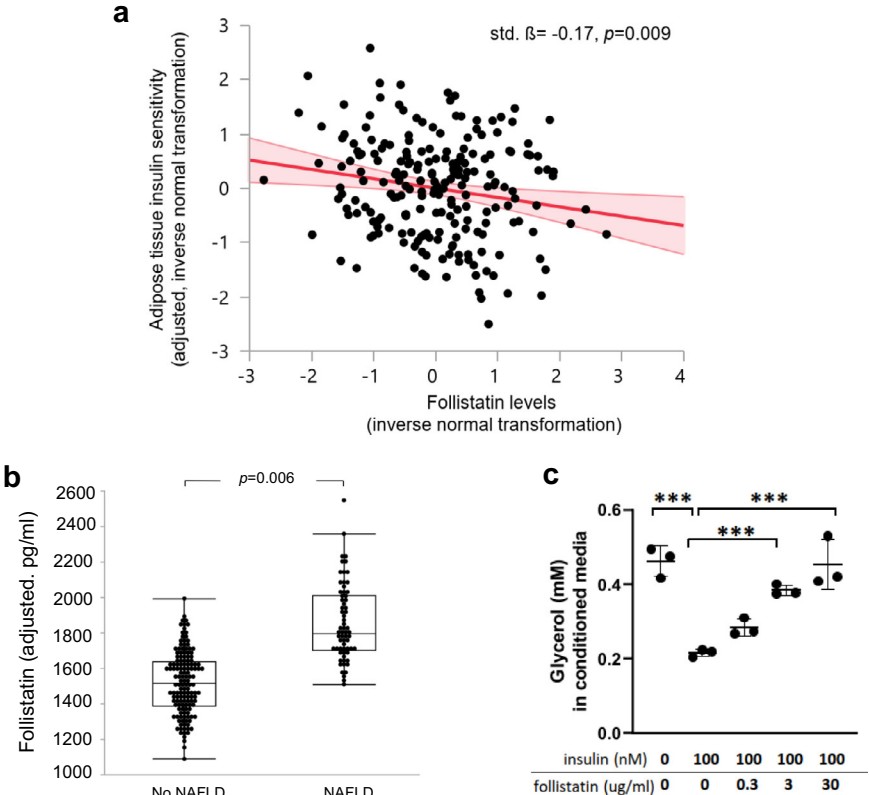

**Fig. 2 Association of follistatin with adipose tissue insulin resistance and nonalcoholic fatty liver disease (NAFLD). a** Relationship of adipose tissue insulin sensitivity, adjusted for age, sex, and total body fat mass, with follistatin levels, in a multivariate linear regression model in the Tübingen Diabetes Family Study (TDFS) cohort ($n = 210$). Depicted is the regression line and the 95% CIs. **b** Relationship of follistatin levels, adjusted for age, sex, and total body fat mass, with NAFLD, in a multivariate logistic regression model (TDFS, $n = 210$). **c** Human adipose-derived stem cells (Lonza) were differentiated into adipocytes (see details in Methods). Cells were maintained in insulin-free media overnight and treated with 0, 0.3, 3, and 30 μg/mL follistatin for 2 h followed by addition of 100 ng/mL insulin for 3 h. Breakdown of triglyceride released glycerol and free fatty acids (FFAs), and lipolysis was determined by measuring media glycerol content. Three independent experiments ($n = 3$) were performed. Statistical significance was determined by Tukey–Kramer HSD using JMP 14.1.0. ***$p < 0.001$ as indicated.

genetic regulators of circulating follistatin levels (summarized in Fig. 5).

In the MDC-CC cohort, elevated plasma follistatin levels associated with the incidence of diabetes up to 19 years before the onset of the disease. While it is important that this association was found during a long period of follow-up, parameters other than elevated follistatin levels, which may become relevant just prior to the onset of diabetes, or also associate with elevated follistatin levels, may contribute to this relationship. Therefore, we analyzed an independent IMI-DIRECT-METSIM cohort, with 4-year follow-up. The independent association between the elevated follistatin and an increased risk of diabetes was also confirmed in this shorter follow-up period. We also included fasting glucose and CRP to the models to ensure that factors in the subclinical phase of diabetes development were not responsible for the raised levels of follistatin. These data suggest that increased circulating follistatin may serve as a marker of diabetes risk, which may be relevant to indicate an increased risk of diabetes, up to 19 years prior to the manifestation of the disease. However, it is worth noting that our C-statistics analysis also suggests that follistatin on its own may not improve diabetes prediction beyond established risk factors.

Previous investigations have presented rather contradictory evidence on the effects of follistatin under physiological and pathophysiological conditions[13–15]. Nevertheless, the physiological regulation of follistatin secretion and pathological effects of abnormally elevated follistatin may represent different avenues. Under normal physiological conditions, disruption of the GCKR-

GCK complex, triggered by e.g., glucose and fructose[22], stimulates follistatin secretion, which is regulated by insulin and glucagon. However, in an insulin-resistant state, attenuated insulin signaling in the liver may lead to elevated follistatin secretion as previously shown in mouse models[15]. Abnormally elevated follistatin secretion may further exacerbate liver insulin resistance by promoting FFA production from adipose tissue and ultimately lead to NAFLD, possibly aggravating diabetes. The previous finding that follistatin increases beta-cell proliferation during normal physiological conditions[13] is perfectly in line with the need for increased insulin secretion to compensate for insulin resistance, and furthermore raises the intriguing possibility that follistatin plays a key role in mediating this signal from the liver to the pancreatic beta cell. Further studies are needed to understand the role of follistatin in the cross-talk between liver and pancreas in normal, as well as pathophysiological conditions.

After investigating mechanisms by which follistatin may be involved in the pathophysiology of T2D, our data suggest that the effect of follistatin in promoting adipose tissue insulin resistance in mice[15] may also be operative in humans. We found that circulating follistatin correlated negatively with adipose tissue insulin sensitivity and positively with circulating FFA levels and liver fat content. Interestingly, the relationship of follistatin with liver fat was affected by adjustment for the percentage of leg fat mass, which suggests that follistatin may impair insulin sensitivity of adipose tissue, predominantly in leg fat, the fat compartment that is considered to be a reservoir for fatty acid storage[23,24]. In

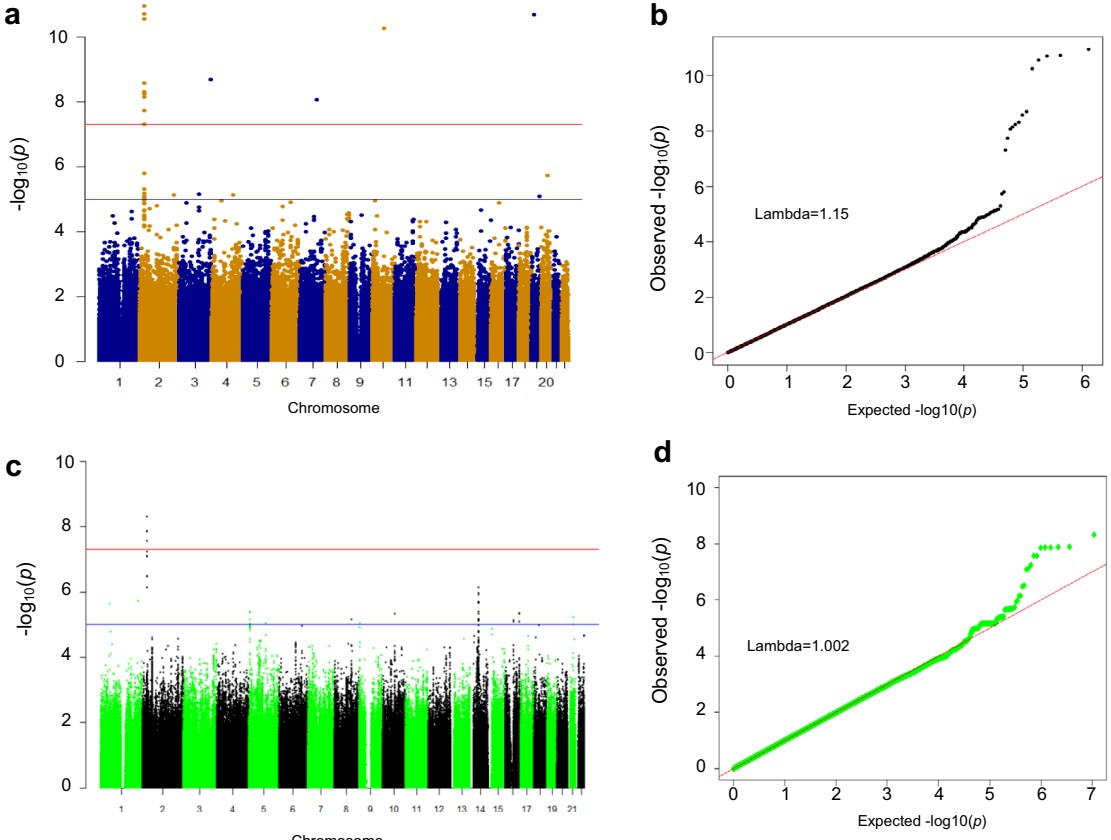

**Fig. 3 GWAS of follistatin in the MDC-CC and SUMMIT-VIP cohort. a** Manhattan plot ($-\log_{10}$ GWAS plot) of GWAS on plasma follistatin in individuals of MDC-CC cohort ($n = 4195$). The most significant SNPs in this analysis are rs780094, rs780093, and rs1260326 in the *GCKR* gene (glucokinase regulatory protein). **b** Quantile-quantile (QQ) plot of the data shown in the Manhattan plot in **a**. **c** Manhattan plot ($-\log_{10}$ GWAS plot) of GWAS on plasma follistatin in individuals of SUMMIT cohort ($n = 885$). The most significant SNP in this analysis is rs1260326 in the *GCKR* gene. **d** Quantile-quantile (QQ) plot of the data shown in the Manhattan plot in **c**. The genome-wide significance level is set at $5 \times 10^{-8}$ and plotted as the dotted line.

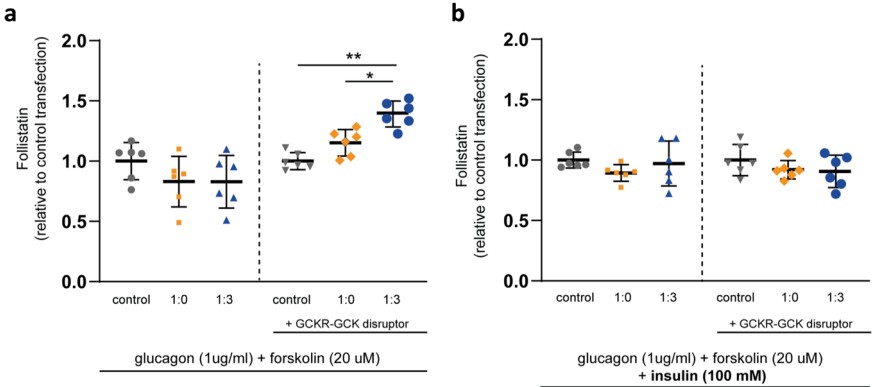

**Fig. 4 Liver cell follistatin secretion is controlled by the glucokinase regulatory protein- glucokinase (GCKR-GCK) complex. a** Human liver carcinoma-derived HepG2 cells were transfected with plasmids as indicated: (i) control (pCMV-XL4, grey round plots); (ii) GCK:GCKR (1:0; no GCKR, orange square plots); (iii) GCK:GCKR (1:3, blue triangle plots). Forty-eight hours after transfection, cells were serum starved in 5.5 mM DMEM for 3 h, and a GCKR-GCK disruptor molecule AMG-3969 (0.7 μM) was added in the medium for 30 min. Cells were then incubated in serum-free low glucose (5.5 mM) DMEM containing glucagon (0.3 μM) and forskolin (20 μM), and AMG-3969 (0.7 μM) was added to respective wells. After 4-hour incubation, the medium was collected for follistatin assay by ELISA. Follistatin levels were normalized to the protein concentration within each sample. **b** HepG2 cells were treated as described in panel **a**, but in the presence of insulin (100 mM). Two independent experiments with 3 technical replicates per condition were performed in different days using different plasmid preparations and cell passage numbers ($n = 6$; *$p = 0.03$ and **$p = 0.005$; ANCOVA with "experiment" as covariable and LSD post-hoc test; data are presented as mean values $+/-$ SD).

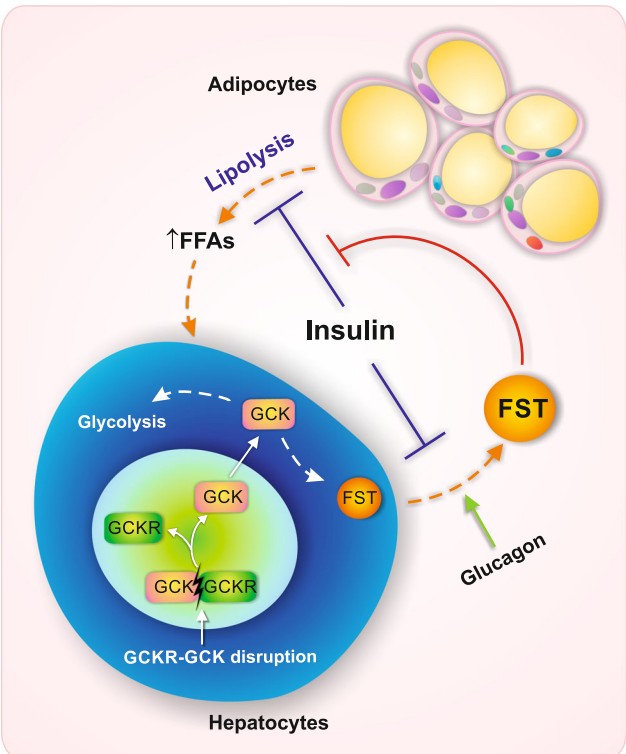

**Fig. 5 Schematic illustration of crosstalk between liver follistatin production and adipocyte insulin-inhibited lipolysis.** Liver follistatin (FST) secretion is regulated by GCKR-GCK, which is stimulated by glucagon and inhibited by insulin. Elevated follistatin may attenuate insulin-inhibited lipolysis in the adipocytes and mediate adipose tissue insulin resistance and free fatty acid (FFA) release, which ultimately contributes to T2D and NAFLD risk.

this respect, it is important to note that in our study adipose tissue insulin sensitivity correlated positively with the percentage of leg fat mass, but negatively with visceral fat mass. Consequently, it could be speculated that follistatin-induced adipose tissue insulin resistance shifts fatty acids not only to the visceral fat depots, but also to the liver and, thereby, aggravates NAFLD. This view was further supported by our in vitro adipocyte data that excess follistatin attenuated insulin-inhibited lipolysis and elevated FFA release, which in turn may predispose to NAFLD. Increased FFAs per se also stimulate glycogenolysis, gluconeogenesis and, thereby, contribute to increased risk of T2D[25]. However, it is not fully understood how follistatin regulates lipolysis in adipocytes. It has been shown in a previous investigation in mice that follistatin suppresses the interaction between insulin receptor substrate 1 (Irs1) and p110a subunit of PI3-kinase in adipose tissue of LDKO mice, which was consistent with reduced AKT activation, attenuated phosphorylation, and inactivation of hormone-sensitive lipase (HSL)[15]. The exact mechanisms by which follistatin inhibits the association between Irs1 and p110a requires further investigation. Furthermore, in our in vitro experiments on human adipocytes, the concentrations of follistatin were relatively higher than those in the serum of healthy subjects during an overnight fast. Nevertheless, it has also been shown that serum follistatin levels may vary considerably in different situations, such as pregnancy and in various disease states[26], as well as under a mixed meal test, exercise, and prolonged fasting[26–28]. We may speculate that under certain conditions higher follistatin concentrations may also be present in vivo.

Besides our previous and present findings on the impact of follistatin on the regulation of glucose and lipid metabolism in animals

and in vitro, to further investigate causal relationships of follistatin with an incidence of diabetes, Mendelian randomization (MR) analysis would be an instrumental approach. Using variants of *GCKR* in MR analysis, which we identified to be most strongly associated with follistatin levels, is nevertheless very problematic, because these variants of *GCKR* were found to be the most pleiotropic variants in our exome sequencing study of close to 10,000 individuals (unpublished data in the DIRECT-METSIM study). In this respect, the variant rs780094 in *GCKR*, which most strongly associated with follistatin levels in our study, was found to associate with elevated fasting and random glycemia and increased risk of T2D in several large studies. However, it also strongly associated with lower CRP, triglyceride, LDL cholesterol levels, and other diabetes risk parameters (https://t2d.hugeamp.org/variant.html?variant=rs780094).

Our GWAS analyses in two independent cohorts identified the SNP rs1260326 in the *GCKR* gene to strongly associate with plasma follistatin levels. This SNP has also been shown in previous studies to be associated with more than 25 metabolic traits, including T2D risk, NAFLD, fasting insulin, total cholesterol, as well as circulating levels of various metabolites[29]. It is noteworthy that functional variants in the *GCKR* gene have been associated with opposite effects on fasting plasma triglyceride and glucose concentrations[16]. Experiments in diabetic animals also showed that the small molecule disruptor of the GCKR-GCK complex AMG-1694 lowered blood glucose, but increased triglyceride levels[21]. Based on our findings, this phenomenon may now be explained by the fact that the GCKR-GCK disruptor may have increased liver cell follistatin secretion, which in turn, may have promoted adipose tissue insulin resistance and an increase of triglyceride levels.

The molecular mechanisms by which the GCK-GCKR system regulates follistatin secretion in hepatic cells is not known. GCKR regulates GCK, playing an important role in regulating the rate of glucose metabolism in the hepatocyte. Thus, we can hypothesize that the association of *GCKR* locus with follistatin levels may be explained by the effects of GCKR regulatory SNPs on GCKR function in the liver. To examine how GCKR-GCK may regulate the secretion of follistatin from the liver, we employed in vitro experiments in the human hepatocyte cell line HepG2. Although the HepG2 cell line shows altered metabolic activity compared to primary hepatocytes, the cell line serves as an acceptable model to study how GCKR-GCK nuclear dissociation and GCK translocation to the cytoplasm regulate follistatin secretion. Transfection of GCK alone did not increase follistatin secretion in HepG2 cells. This may be explained by the very low endogenous expression of GCKR in HepG2 cells, in the absence of which GCK is decreased, likely through degradation[30–32].

In the liver, GCK facilitates the storage of glucose as glycogen, which is under the control of GCKR via binding and subsequently inactivation of GCK in the nucleus of the hepatocyte. Disruption of the GCKR–GCK complex enables the translocation of GCK from the nucleus to the cytoplasm where it facilitates the conversion of glucose to glucose-6-P[33]. In the postprandial state with increased glucose levels, GCKR-GCK complex is disrupted and GCK is released to the cytoplasm where it stimulates glycolysis, glycogen formation, and de novo lipogenesis, while in the fasting state, GCKR suppresses GCK and subsequently hepatic glycolysis. Notably, we provide a mechanism linking glucagon- and insulin-regulated follistatin secretion to GCKR. Here we observed that glucagon was only capable of stimulating follistatin secretion from the liver cells upon GCKR-GCK disruption. This may induce a vicious cycle in that both the high glucose and high glucagon, increase follistatin secretion, resulting in increased adipose tissue lipolysis and circulating FFAs. Furthermore, we also observed that insulin suppresses follistatin secretion independently of glucagon-GCK-GCKR signaling. Nevertheless, as

the GCKR variants are associated with multiple molecular traits, including multiple other protein markers, at this stage, we do not know if the malfunction of GCKR is peculiar to follistatin, or if other proteins are involved.

The *GCKR* gene is in close proximity to *ZNF512*, *C2orfl6*, and *GPN1* on chromosome 2, which also contains SNPs showing significant association with plasma follistatin levels. Other SNPs showing significant association with plasma follistatin include exm1435650 in the *ZNF333* gene ($p = 1.98E-11$), exm831645 in *ADAMTS14* ($p = 5.50E-11$), exm373967 in *TEME44* ($p = 2.05E-09$) and exm646262 in *CUX1* which have been shown to be genetic factors influencing elevated markers of death receptor-activated apoptosis associated with increased diabetes and cardiovascular disease risks[34]. The effect of these identified genes strongly associated with circulating follistatin requires further investigation.

In conclusion, elevated circulating follistatin associates with an increased risk of incident T2D, independently of established diabetes risk markers. Among the mechanisms that may explain this relationship in humans we found supporting evidence that follistatin induces insulin resistance in adipose tissue, thereby, promoting adipose tissue lipolysis and NAFLD. Furthermore, we found that the GCK-GCKR complex may be involved in the regulation of plasma follistatin levels in humans. Thus, follistatin may be an attractive target for therapeutic interventions to prevent T2D and NAFLD.

## Methods

**Study cohorts and biomarker measurements**. Detailed description of study participants, plasma biomarker measurements, phenotyping procedures and GWAS of the Malmö Diet and Cancer study (MDC-CC); the SUrrogate markers for Micro- and Macro-vascular hard endpoints for Innovative diabetes Tools-Vascular Imaging Project (SUMMIT-VIP) study cohort; the Innovative Medicines Initiative Diabetes Research on Patient Stratification (IMI-DIRECT) cohort including its sub-cohort IMI-DIRECT-METSIM; and the Tübingen Diabetes Family Study (TDFS) cohort are presented in the Supplementary Information.

**Follistatin measurements**. Plasma follistatin levels in the MDC-CC, IMI-DIRECT-METSIM, and SUMMIT-VIP cohorts were assessed by Proximity Extension Assay (PEA) in the Olink platform (https://www.olink.com), where different antibody types were validated to ensure reproducible quantifications of the analyte in biological samples[35]. This method is based on a matched pair of antibodies linked to unique oligonucleotides binds to the respective protein target, and DNA amplicon can be subsequently quantified by quantitative real-time PCR. Specifically, follistatin was measured using a Proseek Multiplex CVD I $96 \times 96$ Kit (Olink Bioscience, Uppsala, Sweden) assay based on the Proximity Extension Ligation technology on a Fluidigm BioMark HD real-time PCR platform in 54 chip runs[35]. The lower and upper limits of follistatin were 1.91 and 62,500 pg/mL, respectively. The samples were consecutively aliquoted on plates, regardless of future diabetes status. The plates were analysed in random order. Follistatin concentrations are presented as normalized protein expression (NPX) arbitrary units (AU) with log2 scale calculated from Ct values which was converted into the linear scale ($2^{NPX} =$ linear NPX). The NPX measurement provides relative qualification, and samples can be compared within a cohort but not in two separate cohort studies with different scales as seen in MDC-CC and IMI-DIRECT-METSIM cohorts, where epidemiological findings based on follistatin measurements were robustly reproduced independently. Furthermore, the PEA offers highly specific and sensitive large-scale measurement and has been shown to be comparable to ELISA method[36]. The intra-assay coefficient of variation (CV) was 9%, and the inter-assay CV was 15%. CV was calculated per assay using the assumption of a log-normal distribution. The CV was then averaged across panels (www.olink.com for details). For smaller sample size in the TDFS cohort, follistatin was measured by a human Follistatin ELISA kit (DFN00, R&D Systems).

**Genome-wide association study**
*MDC-CC*

## Genotyping
In MDC-CC, non-fasting blood samples were drawn at the baseline examination and stored in the biobank at −80 °C. All individuals with information on follistatin plasma levels and genotypes ($n = 4239$) were included. Genotyping was performed using the Illumina HumanOmniExpressExome Bead Chip on the iScan system and using the Autocall calling algorithm (Illumina, San Diego, CA, USA). For GWAS, we included all individuals with information on follistatin plasma levels and genotypes ($n = 4239$).

Quality control was performed by exclusion of missingness > 0.05 (both individual and genotypic); identity-by-descent (IBD) match; heterozygosity (absolute cryptic relatedness inbreeding coefficient > 0.2); sex mismatch; population outliers. The minor allele frequency limit was 0.01. A total of 628526 SNPs was included in the analysis.

Linear regression models, with an additive genetic model, were used to test the association between genetic variants and follistatin levels, with adjustment for age and sex. A $p$ value $< 5 \times 10^{-8}$ was considered as genome-wide significant, corresponding to a Bonferroni correction for one million tests. Version 1.07 of PLINK software was used for association analyses and QC. Manhattan plots and Q-Q plots were drawn with the R software version 3.1.2. Regional significance plots were drawn using LocusZoom (http://locuszoom.sph.umich.edu/locuszoom/).

## SUMMIT
Raw sequencing data were subjected to quality control and imputation using an array of well-developed and optimized software before the data analysis started. For quality control samples were filtered based on individual characteristics; genotyping rate, sex check, population stratification, identified by descent and heterozygosity and variant sequencing quality; genotyping rate, Hardy–Weinberg equilibrium, minor allele frequency, and minor allele count. QC Protocols using PLINK produced efficient and reproducible QC that can be customized to suit different genotype datasets. Quality control filtering cleaned the data from individuals that can cause errors in the analysis; duplicated individual samples, first-degree relatives, individuals from different ancestry, and individuals with low genotyping rate. Rare variants are those single-nucleotide polymorphisms (SNPs) that deviate significantly from Hardy–Weinberg equilibrium and can cause false positives.

SUMMIT dataset was imputed by HRC Michigan Imputation Server. Haplotype Reference Consortium (HRC) is a large reference panels that has 64970 human haplotypes and 39,235,157 snps from 20 studies mainly from European ancestry. HRC server offers an imputation platform for genotype data. Genotype data that passed quality control were separated to chromosome files, each file has the SNPs of a single chromosome to be imputed and they are usually in VCF format. The imputed chromosome files along with a quality report were downloaded from the server.

GWAS was performed for the HRC imputed SUMMIT dataset using SNPTEST (snptest version 2.5.2). The SNPTEST output was filtered for the minor allele frequency (MAF > 0.05), INFO score (> 0.4), and Hardy–Weinberg equilibrium (HWE > 0.0000057). Manhattan plots and Q-Q plots were generated in R (R version 3.4.3).

**Human adipocyte differentiation and lipolysis measurement**. Human adipose-derived stem cells were obtained from Lonza and seeded in 96-well plates (160,000 cells/cm$^2$) in 100 µL EGM2-MV medium (Zenbio). After 24 h, the medium was replaced with PM1 (Zenbio) for 24 h, followed by initiation of differentiation with DM2 (Zenbio) medium for 6 days. Subsequently, the cells were maintained in AM1 (Zenbio) every 3-days until assays were performed. Differentiated adipocytes were utilized between days 12 and 14 postinitiation of differentiation. For lipolysis assays, differentiated adipocytes were maintained in insulin-free medium (PM1) overnight in DMEM (11054-020, Gibco) containing 0.2% fatty acid-free BSA (03117057001, Roche) with 0, 0.3, 3, or 30 µg/mL follistatin (120-13, PeproTech) for 2 h followed by addition of 100 ng/mL insulin (SLBX8532, Sigma) for 3 h. Lipolysis was determined by measuring media glycerol content (F6428, Sigma). Data were plotted using GraphPad Prism and significance was determined using Dunnett's Test in JMP.

**Follistatin secretion in HEPG2 cells**. Human liver carcinoma-derived HepG2 cells (ATCC, HB-8065) were cultured in 5.5 mM Dulbecco's modified Eagle medium (DMEM) supplemented with 2 mM L-glutamine, 100 U/ml penicillin,100 µg/ml streptomycin (LONZA), and 10% fetal bovine serum (GIBCO). Cells were seeded on 24 well plates and incubated overnight at 37 °C in 5% CO$_2$, and then transfected with plasmids using lipofectamine 3000 according to manufacturer´s instructions (Thermo Fisher Scientific). Forty-eight hours after transfection, cells were serum starved in 5.5 mM DMEM for 3 h and AMG-3969 (0.7 µM, MCE) was added for 30 min. Next, cells were incubated with or without insulin (100 mM) together with glucagon (0.3 µM) and intracellular cAMP activator forskolin (20 µM) in low glucose DMEM medium (5.5 mM), conditions previously shown to stimulate follistatin secretion in liver cells[13]. AMG-3969 was added to respective wells. Media was collected after 4 h and diluted (1:2) for follistatin ELISA assay (DFN00, R&D Systems) according to manufacturer's instructions.

**Statistics**. In MDC-CC and IMI-DIRECT-METSIM analyses, one-way analysis of variance for continuous variables and Pearson's Chi-squared test for dichotomous variables were used to assess the cross-sectional relationships between plasma follistatin quartiles and diabetes risk factors. Multiple linear regression was used to analyze the association between follistatin and glucose, HbA$_{1c}$, HOMA2, and insulin at baseline and reexamination, adjusted for potential confounding factors. Natural log-transformed values for HOMA2, insulin and CRP were used due to skewed distributions.

Cox proportional hazards regression was used to examine hazard ratios (HR) with 95% confidence interval (CI) for incidence of diabetes, by quartiles of

follistatin and per 1 standard deviation (SD) increment, using the lowest quartile as the reference category. Potential confounders were age, sex, waist circumference, smoking habits, LDL, HDL cholesterol, fasting glucose, systolic blood pressure, antihypertensive medications, lipid-lowering medications, CRP, BMI, physical activity, alcohol intake, and fiber intake. The fit of the proportional hazards model was confirmed by plotting the incidence rate over time. The Kaplan–Meier curve was used to illustrate the incidence of diabetes in relation to the follistatin quartiles. Time axis was follow-up time until death, emigration, incident diabetes or end of follow-up. SPSS Statistics (version 22) and Stata software version 12.0 (Stata Corp, College Station, TX, USA) were used for statistical analyses.

In TDFS cohort analysis data are given as mean ± SD. For statistical analyses data that were not normally distributed (Shapiro-Wilk W Test) were inverse normal transformed to achieve a normal distribution for the investigation of univariate and multivariate relationships. Pearson correlation and the nonparametric Wilcoxon test were used to investigate univariate relationships between the parameters. Multivariate models were used to investigate independent relationships. The statistical software package JMP 13.0.0 (SAS Institute Inc, Cary, North Carolina) was used.

**Ethics**. The study was performed in accordance with the Declaration of Helsinki. The ethics committee at Lund University approved the MDC; and SUMMIT study was approved by ethics committees in each of the centers in Malmö (Sweden), Pisa (Italy), Dundee (U.K.), and Exeter (U.K.). For DIRECT-METSIM cohort, approval for the study protocol was obtained from the Ethics Committee of the University of Eastern Finland and Kuopio University Hospital and all participants provided written informed consent at enrolment. The research conformed to the ethical principles for medical research involving human participants outlined in the declaration of Helsinki. For TDFS cohort, informed written consent was obtained from all participants and the Medical Ethics Committee of the University of Tübingen had approved the protocol.

**Reporting Summary**. Further information on research design is available in the Nature Research Reporting Summary linked to this article.

## Data availability

All the relevant data supporting the findings of this study are available within this article, in the supplementary material, the source data file, or relevant repositories. For MDC-CC, SUMMIT and TDFS cohorts, Swedish, European and German legislation impose restrictions on public availability of datasets containing pseudonymized information. The full datasets including genome-wide data and phenotypes can be accessed for the MDC-CC through an institutional repository at Lund University (https://www.malmo-kohorter.lu.se/english), and SUMMIT through the SUMMIT vascular imaging project steering committee (jan.nilsson@med.lu.se), and University of Tübingen (Norbert.Stefan@med.uni-tuebingen.de) with pertinent permissions. Details on how to requests access to IMI-DIRECT data, including data presented here, can be found through https://directdiabetes.org/contacts/. Requestors will be provided with information and assistance on how data can be accessed via the DIRECT Computerome following submission of appropriate documentation. Source data are provided with this paper.

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

## Acknowledgements

This study was supported by the Swedish Research Council, Strategic Research Area Exodiab, Dnr 2009-1039; and the Swedish Foundation for Strategic Research Dnr IRC15-0067. The authors acknowledge the information provided by the National Diabetes Register

of Sweden, the Malmö HbA$_{1c}$ register, the Diabetes 2000 registry, and the registers provided by the Swedish Board of Health and Welfare. The research leading to these results has received funding from the European Community's Seventh Framework Programme (FP7/ 2007-2013) for the Innovative Medicines Initiative under grant agreement n° 115006, the SUMMIT consortium. Furthermore, the research was supported with a project grant from the Swedish Research Council to LG (2015-02558), and a European Foundation for the Study of Diabetes (EFSD) grant and Hjelt Foundation grant to YDM. The study was also funded by the Swedish Heart–Lung Foundation (Grant No. 2017-0626, 20170520 and 20180640), the Swedish Research Council (VR 201502811, 2017-02688), the Novo Nordisk Foundation (NNF18OC0034408), and Diabetes Wellness Sweden (25-420 PG). This project has received funding from the Innovative Medicines Initiative 2 Joint Undertaking under grant agreement No 115974 (BEAt-DKD). This Joint Undertaking receives support from the European Union's Horizon 2020 research and innovation programme and EFPIA with JDRF. The IMI-DIRECT cohorts were supported by the Innovative Medicines Initiative Joint Undertaking under grant agreement n°115317 (DIRECT), resources of which are composed of financial contribution from the European Union's Seventh Framework Programme (FP7/2007-2013) and EFPIA companies' in-kind contribution. This project has received funding from the Innovative Medicines Initiative 2 Joint Undertaking under grant agreement No 115881 (RHAPSODY). This Joint Undertaking receives support from the European Union's Horizon 2020 research and innovation programme and EFPIA. This work is supported by the Swiss State Secretariat for Education, Research and Innovation (SERI) under contract number 16.0097. The opinions expressed and arguments employed herein do not necessarily reflect the official views of these funding bodies. Financial support has been given to the Malmö Diet and Cancer Study by the Swedish Cancer Foundation, the Swedish Medical Research Council, the European Commission, the City of Malmö, the Swedish Dairy Association and the Albert Påhlsson Foundation. We also acknowledge support from a Lund University Infrastructure grant: "Malmö Population-Based cohorts" (STYR 2019/2046). This study included data from register linkage with the National Diabetes Register (NDR) of Sweden, a register containing data from both hospital care and primary health care for patients with diabetes. CW and RG were supported by the National Natural Scientific Foundation of China (NSFC) under Grant U1806202 and 61533011. ACS was supported by the NIHR Exeter Clinical Research Facility; the views expressed here are those of the authors and not those of the UK Department of Health or NIHR. NS is currently supported by a Heisenberg-Professorship of the Deutsche For-schungsgemeinschaft (STE 1096/3-1) and a Visiting Professorship from the Harvard Medical School. European Research Council (ERC) under the European Union's Horizon 2020 research and innovation program (Grant No. 802825 to M.U.K) and Jane and Aatos Erkko Foundation (MUK). We would like to thank the Clinical Biomarker Facility at SciLifeLab, Sweden, for providing assistance in protein analyses; and Matilda Dale and Dr. Ragna Häussler for their work on Olink for the IMI-DIRECT. We also thank AstraZeneca for suggestions on implementation of the research.

## Author contributions

C.W., Y.B., E.A., C.L., M.G.H., J.M.S., D.M.A., M.C., and R.W. performed data analysis on cohorts. M.L.R., W.C.R., J.M.W., A.R., performed in vitro experiments. A.P., J.M., H.S., A.F., R.T., P.N., A.C.S., F.K., A.N., O.M., M.O.M, J.N., H.U.H., E.R.P, P.W.F., A.L.B., and N.S. provided cohorts to the study. R.G., M.U.K., G.E., J.W., E.R., C.B.W., M.F.W., K.L.D., A.A.V., M.L., N.S., L.G., and Y.D.M. contributed to the implementation of the research and the final version of the manuscript. M.L., N.S., L.G., and Y.D.M. designed and supervised the study. Y.D.M. was in charge of overall direction and planning, and wrote the paper with input from all authors. All authors reviewed the final manuscript.

## Funding

## Competing interests

W.C.R., J.M.W., A.R., and K.L.D. are employees at Eli Lilly. A patent application has been filed on follistatin and diabetes risk by YDM. The other authors declare no competing interests.

## Additional information

[1]Department of Clinical Sciences, Lund University, Malmö, Sweden. [2]School of Control Science and Engineering, Shandong University, Jinan, Shandong, China. [3]School of Intelligent Engineering, Shandong Management University, Jinan, Shandong, China. [4]Institute of Clinical Medicine, Internal Medicine, University of Eastern Finland, Kuopio, Finland. [5]A.I. Virtanen Institute for Molecular Sciences, Department of Biotechnology and Molecular Medicine, University of Eastern Finland, Kuopio, Finland. [6]Lilly Research Laboratories, Eli Lilly and Company, Indianapolis, IN, USA. [7]Department of Internal Medicine IV, Division of Endocrinology, Diabetology and Nephrology; and Department for Diagnostic Laboratory Medicine, Institute for Clinical Chemistry and Pathobiochemistry, University Hospital Tübingen, University of Tübingen, Tübingen, Germany. [8]Institute of Diabetes Research and Metabolic Diseases (IDM) of the Helmholtz Center Munich, Tübingen, Germany. [9]German Center for Diabetes Research (DZD), Tübingen, Germany. [10]Section of Experimental Radiology, Department of Radiology, University of Tübingen, Tübingen, Germany. [11]Division of Endocrinology, Boston Children's Hospital, Harvard Medical School, Boston, MA, USA. [12]Inserm U1283 / CNRS UMR 8199, European Genomic Institute for Diabetes (EGID), Institut Pasteur de Lille; University of Lille, Lille University Hospital, Lille, France. [13]Affinity Proteomics, Science for Life Laboratory, KTH Royal Institute of Technology, Stockholm, Sweden. [14]NIHR Exeter Clinical Research Facility, Royal Devon and Exeter Hospital and University of Exeter Medical School, Exeter, Devon, UK. [15]Division of Systems Medicine, University of Dundee, Ninewells Hospital & Medical School, Dundee, UK. [16]Department of Clinical and Experimental Medicine, University of Pisa, Pisa, Italy. [17]Department of Cell Physiology and Metabolism, University Medical Centre, Geneva, Switzerland. [18]Department of Endocrinology and Metabolism, Division of Life Sciences of Medicine, University of Science and Technology of China, Hefei, China. [19]Division of Population Health & Genomics, School of Medicine, University of Dundee, Dundee DD1 9SY, UK. [20]Steno Diabetes Center Copenhagen, Gentofte, Denmark. [21]Department of Medicine, Kuopio University Hospital, Kuopio, Finland. [22]Finnish Institute for Molecular Medicine, University of Helsinki, Helsinki, Finland. [23]These authors contributed equally: Chuanyan Wu and Yan Borné. [24]On behalf of the IMI-DIRECT consortium. ✉email: yang.de_marinis@med.lu.se

