## [Peer Review File · Nature Communications]

Elevated circulating follistatin associates with an increased risk of type 2 diabetesREVIEWER COMMENTS

Reviewer #1 (Remarks to the Author):

The authors investigated whether circulating follistatin associates with increased risk of incident T2D and dissected the mechanisms involved in its regulation and effects on T2D. Elevated circulating follistatin associated with increased risk of incident T2D after adjustment for multiple risk factors in two longitudinal cohorts. High circulating follistatin also associated with adipose tissue insulin sensitivity and non-alcoholic fatty liver disease. In vitro experiments revealed that follistatin dose-dependently increased free fatty acid (FFA) release in human adipocytes. GWAS for circulating follistatin in two independent cohorts identified variation in the glucokinase regulatory protein gene (GCKR) to associate with plasma follistatin levels. Mechanistically, GCKR regulated follistatin secretion in addition to insulin and glucagon in HepG2 cells. The authors concluded that elevated circulating follistatin associates with an increased risk of T2D, at least in part, through induction of adipose tissue insulin resistance.

This reviewer has several specific comments.

1. What is the mechanism by which follistatin regulated lipolysis in adipocytes? Does follistatin directly act on adipocytes? Is it possible that follistatin affects the action of TGF- β family members? Were the doses of follistatin in Figure 2C within physiological ranges?
2. Is adipose tissue the main target organ of follistatin? Is it possible that follistatin modulates insulin sensitivity of skeletal muscle and liver?
3. How does the GCKR-GCK system regulate follistatin secretion in hepatic cells? What is the molecular mechanism? Does the same phenomenon occur in primary hepatocytes? In Figure 4, transfection efficiency of GCKR or GCK should be examined. The authors should show that AMG-3669 effectively promoted translocation of disassociated GCK from the nucleus to the cytoplasm in HepG2 cells.

Reviewer #2 (Remarks to the Author):

The authors report interesting and overall convincing data linking follistatin, a hepatokine, to type 2 diabetes. Strength of the manuscript are the investigation of follistatin in longitudinal cohorts with replication of findings, cross-sectional data with high degree of human phenotyping, GWAS to identify genetic variants, and experimental investigations related to the role of follistatin for FFA release from adipose tissue and the confirmation that the glucose kinase regulator plays an important role for hepatic follistatin secretion. This mix of complementary investigations make the paper a strong and highly informative one.

Major points

- The analysis of different follow-up times is a bit irritating. One issue is of course that different cohorts have different follow-up length. Still, regular Cox regression would assume the proportional hazards assumption is met. If a time-varying effect is hypothesized (this remains unclear to the reviewer), this could be formally addressed in the analyses.
- While the authors adjust for a variety of risk factors, several established risk factors for type 2 diabetes have not been considered. E.g. alcohol consumption, physical activity and diet as well as body height are established risk factors. Given the relatively strong correlation of follistatin with other risk factors, such as BMI, waist circumference, smoking there is considerable risk of confounding bias.
- A major hypothesized pathway relates to the release of FFA from adipose tissue. However, it would be important to clearly demonstrate this relation in humans. While Suppl. Tables show correlations with FFA, these are univariate analyses, not adjusted for other factors.
- GWAS: the results of GWAS indicate the GCKR gene as a prominent locus. However, in contrast to the reporting, the strongest association is not observed for variants in this gene – rather the associations show the lowest p-value. Stronger associations have been observed for variants with lower MAF, e.g. ZNF333, ADAMTS14 or TMEM44. Although such variants are briefly discussed, it

remains unclear why those genes were not followed further. It also remains unclear if the analysis in SUMMIT refers to another exploratory GWAS or a true replication of those variants identified in MDC-CC, the latter would be preferred.

- The authors miss to investigate the causal nature of the diabetes association using Mendelian randomization. This would be a logical step following the GWAS analysis. Importantly in this context, the identified GCKR gene is a major locus for triglyceride levels and also CRP. Thus potential pleiotropic effects versus mediators would need to be carefully considered.

Minor points:

- Abstract: The reporting of associations across different time spans of follow-up is irritating. It is unclear if this reflects a hypothesis of time varying strength of association or merely the fact that different cohorts were used.

- The c-statistics (page 7) indicate that follistatin does not improve diabetes prediction beyond established risk factors (model 2). This should be made clear from the wording in the results and also in the interpretation of the finding in the discussion (page 12). Also, it remains unclear why not measures more suitable for survival data have been used in this context – a c-statistic from logistic regression doesn't account for varying time under risk of participants.

- It would be worthwhile to discuss findings for CRP in more detail. The correlation of follistatin with CRP seems quite substantial. While the authors adjust for CRP levels it would be informative to reflect if CRP needs to be seen as confounder or rather a mediator for a diabetes association. This seems also relevant in light of GCKR identified by GWAS, being also a major locus for CRP.

Reviewer #3 (Remarks to the Author):

This is a very interesting analysis that aims to elucidate the relationship between follistatin and type 2 diabetes risk, as well as mechanisms underlying the associations of interest. The reviewer found this work interesting and of importance.

Specific comments:

1. The introduction left a mixed impression of the function of follistatin under physiological and pathophysiological conditions. For example, the authors suggested that follistatin may improve beta cell mass and counteract glucagon, whereas in other places the language suggested that this cytokine induces insulin resistance. Can the authors address this mixed functions of follistatin? Is it possible that, like leptin, follistatin may function as a marker of various physiological statuses?

2. Line 160-163. The association between follistatin and liver fat contents was not independent of visceral fat mass and leg fat mass. How could the authors suggest that the marker was independently associated with fatty liver in humans?

3. Given that the authors also found significant genetic variants that predict the levels of follistatin levels, a logic next step is to perform Mendelian randomization to explore whether the epidemiological associations are likely causal or not in the MDC-CC cohort. This will make this work much more convincing.

4. Were the CV estimates based on blind QC samples? Need to clearly indicate how the CVs were estimated.

5. The results suggested that the proportional hazards assumption may not hold true. Have the authors examined this assumption in their analysis? It would be informative to plot HR by follow-up duration to understand the temporal pattern of this association.

6. Table 2. A few important variables were not considered in the multivariate adjustment, including physical activity, alcohol intake, BMI as an indicator of overall obesity, and diet quality.

7. Table 4 and Figure 2. The multivariate adjustment is minimal in these analyses.

Elevated circulating follistatin associates with an increased risk of type 2 diabetes

Manuscript ID NCOMMS-20-42000

Response to Reviewers

REVIEWER COMMENTS

Reviewer #1 (Remarks to the Author):

The authors investigated whether circulating follistatin associates with increased risk of incident T2D and dissected the mechanisms involved in its regulation and effects on T2D. Elevated circulating follistatin associated with increased risk of incident T2D after adjustment for multiple risk factors in two longitudinal cohorts. High circulating follistatin also associated with adipose tissue insulin sensitivity and non-alcoholic fatty liver disease. In vitro experiments revealed that follistatin dose-dependently increased free fatty acid (FFA) release in human adipocytes. GWAS for circulating follistatin in two independent cohorts identified variation in the glucokinase regulatory protein gene (GCKR) to associate with plasma follistatin levels. Mechanistically, GCKR regulated follistatin secretion in addition to insulin and glucagon in HepG2 cells. The authors concluded that elevated circulating follistatin associates with an increased risk of T2D, at least in part, through induction of adipose tissue insulin resistance.

This reviewer has several specific comments.

1. What is the mechanism by which follistatin regulated lipolysis in adipocytes?

Answer: We thank the Reviewer for this question. The complete mechanism on how follistatin regulates lipolysis in adipocytes is not fully understood. It has been shown in a previous investigation in mice (*Tao et al., Inactivating hepatic follistatin alleviates hyperglycemia, Nature Medicine 2018*) that follistatin suppresses the interaction between insulin receptor substrate 1 (Irs1) and p110a subunit of PI3-kinase in adipose tissue of LDKO mice, which

is consistent with reduced AKT activation and reduced phosphorylation and inactivation of hormone-sensitive lipase (HSL). However, we do not know exactly how follistatin inhibits the association between Irs1 and p110 α , the elucidation of which requires further investigation. We have now added this information in the “Discussion” (page 14, first paragraph, highlighted in red).

2. Does follistatin directly act on adipocytes?

Answer: Our *in vitro* experiment in Figure 2C (page 9 and 32) showed that follistatin has a direct effect on human adipocytes. In this experiment, human adipocyte-derived stem cells were differentiated into adipocytes, and treated with 0, 0.3, 3 or 30 $\mu\text{g}/\text{mL}$ follistatin for 2 hours before exposure to insulin for 3 hours. Insulin-inhibited lipolysis was attenuated by follistatin dose-dependently.

Although the exact mechanism of the action of follistatin on adipocytes is not known, based on the study we referred to in the response to Question 1 (*Tao et al., Inactivating hepatic follistatin alleviates hyperglycemia, Nature Medicine 2018*), we speculate that follistatin may act directly on adipocytes. In the referred study, 3T3-L1 adipocytes were treated with serum from overnight-fasted 4-month-old LDKO or control mice, with or without the addition of a follistatin antibody (10 $\mu\text{g}/\text{ml}$, AF669, R&D). The results showed that addition of follistatin antibody improved Irs1-p110 α association in the adipocytes. This observation may suggest that follistatin acts on adipose.

3. Is it possible that follistatin affects the action of TGF- β family members?

Answer: We appreciated this comment, and agree that this is an attractive idea. We tested the effect of multiple TGF- β receptor ligands on 3T3L1 cells, but did not observe any effect. We cannot, however, exclude the possibility that we might have missed an important ligand that is inactivated by follistatin on adipose cells.

4. Were the doses of follistatin in Figure 2C within physiological ranges?

Answer: In a previous study on follistatin levels in sera (*Sakamoto et al., Determination of free follistatin levels in sera of normal subjects and patients*

with various diseases. Eur J Endocrinol. 1996 Sep), circulating follistatin concentration was shown to vary during pregnancy and in various diseased states (up to 0.02 ug/ml). The concentrations that we used in the adipocyte *in vitro* experiment are 0.3-30 ug/ml. It's worth to note that tissue local concentration and circulating concentration may differ significantly. This is to be further explored in follow-up studies.

We have now addressed this point in the "Discussion", on page 14, paragraph 1, which now reads:

"However, the exact mechanisms by which follistatin inhibits the association between Irs1 and p110a requires further investigation. **Furthermore, in our *in vitro* experiment on human adipocytes, the concentrations of follistatin we used were relatively higher than those in the serum of healthy subjects during an overnight fast. Nevertheless, it has also been shown that serum follistatin levels may vary considerably in different situations, such as pregnancy and in various disease states²⁶, as well as under a mixed meal test, exercise and prolonged fasting²⁶⁻²⁸. We may speculate that under certain conditions higher follistatin concentrations may also be present *in vivo*.**"

5. Is adipose tissue the main target organ of follistatin? Is it possible that follistatin modulates insulin sensitivity of skeletal muscle and liver?

Answer: Based on the study we referred to in the response to Question 1 (*Tao et al., Inactivating hepatic follistatin alleviates hyperglycemia, Nature Medicine 2018*), we think eWAT is the main target in mice. In the referred study, five-week-old C57BL6 mice were challenged with a HFD for 4 months before infection with Fst315AAV-TBG (liver specific) or GFPAAV-TBG (control), and hyperinsulinemic-euglycemic clamps were performed 4 weeks later. Uptake of 2-DOG tracer into eWAT, iWAT, skeletal muscle and BAT measured at the end of the clamp showed that the effect of follistatin was strongest upon eWAT and weakest upon BAT. Skeletal muscle displayed a weak effect. However, there is no strong conclusion about an effect upon liver. The cause of muscle and BAT insulin resistance in the LDKO mice appear to be through circulating Fgf21 rather than follistatin as shown in a recent study, where Fgf21 does not have a detectable effect on WAT (*Stöhr et al., FoxO1 suppresses Fgf21 during hepatic insulin resistance to impair peripheral glucose utilization and acute cold tolerance. Cell Reports, in press*). This

observation may further support our conclusion that the WAT is the main target of follistatin in the LDKO mice.

6. How does the GCKR-GCK system regulate follistatin secretion in hepatic cells? What is the molecular mechanism? Does the same phenomenon occur in primary hepatocytes?

Answer: The molecular mechanisms by which the GCK-GCKR system regulates follistatin secretion in hepatic cells is not known. GCKR regulates GCK, playing an important role in regulating the rate of glucose metabolism in the hepatocyte. Thus, we can hypothesize that the association of *GCKR* locus with follistatin levels may be explained by the effects of GCKR regulatory SNPs on GCKR function in the liver. This information is now added in “Discussion” (page 15, paragraph 2).

The two main drivers for GCKR pleiotropic associations are rs780094 and rs1260326 *GCKR* SNPs, which are also the two main association signals for follistatin in our study. Functional characterizations of these regulatory loci have demonstrated that rs780094 defines a transcriptional enhancer that regulates *GCKR* expression in an allele specific manner (*López Rodríguez et al., Genome Medicine. 201; 9:63*), whereas the non-synonymous rs1260326 encodes for a proline-by-leucine substitution in GCKR (P446L-GCKR) with an impaired capacity to inhibit GCK (*Rees et al., Diabetologia. 2012; 55:114-22*). Consequently, several authors have proposed that GCKR associations can be explained, at least partially, by the combined effects of these SNPs in GCK/GCKR: a differential GCKR expression (the effect of the enhancer variant rs780094) together with an enhanced GCK activity as a result of a weaker inhibition by P446L-GCKR.

A recent study has demonstrated that P446L-GCKR induces a NADH reductive stress that may functionally mediate the association of this variant with several metabolic traits (*Goodman et al., Nature. 2020; 583:122–126*). Interestingly, our unpublished RNA-seq data from liver cell lines show that follistatin expression is strongly affected by changes in GCK/GCKR. Whether follistatin is regulated by the NADH reductive stress and/or by GCKR-GCK mediated effects on gene expression should be further investigated.

7. In Figure 4, transfection efficiency of GCKR or GCK should be examined.

Answer: We thank the reviewer for this comment and bringing this topic out. We have optimized and validated the transfection of HepG2 cells with Lipofectamine 3000 reagent and we estimate the efficiency to be above 65%. Specifically, data showing the efficiency of transfection with the same expressing plasmids used in our study have been previously published (*López Rodríguez et al., Scientific Reports volume 8, Article number: 15989, 2018*). In addition, we have demonstrated high transfection efficiency in HepG2 cells with other plasmids in a different study (*López Rodríguez et al., Genome Medicine volume 9, Article number: 63 2017*). Here we also demonstrated high transfection efficiency using a GFP-GCKR plasmid. This information is now added in the manuscript as Figure S3 (page 11 and 60).

Figure S3

Figure S3. HepG2 cells were transfected with GFP-GCKR (RG214230, Origene) using Lipofectamine 3000 (Thermo Fisher Scientific) according to manufacturer instructions. The cells were imaged 48 h post-transfection in a Zeiss AXIO Observer. Z1 microscope controlled by Zen imaging software (Zeiss). Both fluorescent (left picture) and bright-field (right panel) images were acquired.

8. The authors should show that AMG-3669 effectively promoted translocation of disassociated GCK from the nucleus to the cytoplasm in HepG2 cells.

Answer: We thank the reviewer for this comment that prompted us to verify this aspect of our *in vitro* experiments. AMG-3969, and its chemical precursor

AMG-1694, were selected for their capacity to disrupt GCK-GCKR binding in an *in vitro* screening of a small molecule library. Their effect on GCK cytoplasmic translocation was verified in the original report and was the base for the functional characterization (Lloyd *et al.*, *Nature*. 2013 Dec 19;504(7480):437-40. doi: 10.1038/nature12724).

We now verified the effect of AMG-3969 on GCK translocation in our overexpression system. Addition of AMG-3969 (GCK-GCKR +) induced GCK translocation and produced similar localization pattern to that of free GCK. Nuclear localization of GCK is highest in the absence of the disruptor (GCK-GCKR -). The data is now added in the Results (page 11), Figure S4, S5 (page 61-62), and Supplementary Material (page 45-46). Figure S4 and S5 are highlighted below.

Figure S4

Figure S4. Cellular localization of transfected GCK on fixed HepG2 cells as visualized by fluorescence imaging using a primary polyclonal antibody for human GCK (ab88056). Green fluorescent images (left panels, Alexa Fluor 488) and overlay images (right panels, composite Alexa Fluor 488 + DAPI [nuclear dye]) are shown. Arrows depict cytoplasmic or nuclear localization. Lower right panels are zoomed-in of the selected area in the original image.

Figure S5

Figure S5. Violin plots showing the distribution density for the Nuclear % of the Area, Perimeter and Integrated density in each treatment.

Reviewer #2 (Remarks to the Author):

The authors report interesting and overall convincing data linking follistatin, a hepatokine, to type 2 diabetes. Strength of the manuscript are the investigation of follistatin in longitudinal cohorts with replication of findings, cross-sectional data with high degree of human phenotyping, GWAS to identify genetic variants, and experimental investigations related to the role of follistatin for FFA release from adipose tissue and the confirmation that the glucose kinase regulator plays an important role for hepatic follistatin secretion. This mix of complementary investigations make the paper a strong and highly informative one.

Major points

1. The analysis of different follow-up times is a bit irritating. One issue is of course that different cohorts have different follow-up length. Still, regular Cox regression would assume the proportional hazards assumption is met. If a time-varying effect is hypothesized (this remains unclear to the reviewer), this could be formally addressed in the analyses.

Answer: We thank the Reviewer for this comment. The rationale of replicating association studies in two different cohorts with different follow-up periods is due to the limited information on fluctuations of follistatin over time, or the impact of other conditions on follistatin levels (i.e., infections etc.). The follow-up period for the MDC-CC cohort was 19.07 years, and the duration of time from follistatin measurement and diabetes diagnosis is rather long. To verify that the measurement of follistatin at a single point in time is reflective of true exposure, we performed analyses for incidence of diabetes in an independent cohort DIRECT-METSIM with follow-up of 4 years. We also agree with the Reviewer that associations across different time spans of follow-up within the same MDC-CC cohort may be rather confusing and have now removed the 5-year association analysis in MDC. The current analyses include MDC-CC (19-year) and replication in DIRECT-METSIM (4-year), by which we address both time varying strength of association and replication in different cohorts. The changes are highlighted in “**Abstract**” (page 4), and “**Results**” under heading “PLASMA FOLLISTATIN LEVELS ASSOCIATE WITH THE RISK OF T2D” (page 7-8).

2. While the authors adjust for a variety of risk factors, several established risk factors for type 2 diabetes have not been considered. E.g. alcohol consumption, physical activity and diet as well as body height are established risk factors. Given the relatively strong correlation of follistatin with other risk factors, such as BMI, waist circumference, smoking there is considerable risk of confounding bias.

Answer: We appreciate this comment from the Reviewer, which is a very relevant point. We have now analyzed the incidence of diabetes in relation to sex-specific quartiles of plasma follistatin levels in MDC-CC in the following three models: **Model 1: Adjusted for age and sex. Model 2: Adjusted for age, sex, BMI, physical activity, alcohol intake, fiber intake, smoking habits, use of anti-hypertensive medications, systolic blood pressure, LDL, HDL cholesterol, use of lipid lowering medications, fasting glucose. Model 3: Model 2 and CRP.** HR per SD increase in follistatin levels for T2D is 1.29 (CI: 1.19-1.40, $p<0.001$), adjusted for age and sex (Model 1); 1.31 (CI: 1.11-1.56, $p<0.01$) in Model 2, and 1.24 (CI: 1.04-1.47, $p<0.05$) in Model 3 (“Results”, page 7; Table 2, page 28). Accordingly, we also updated the C-statistics analysis with additional variables (page 7, last paragraph).

We also performed the same analysis for the replication cohort DIRECT-METSIM. HR per SD increase in follistatin levels for T2D is 1.35 (CI: 1.13-1.61, $p<0.01$) in Model 1 adjusted for age; 1.28 (CI: 1.075-1.524, $p<0.01$) in Model 2 adjusted for age, BMI, physical activity, alcohol intake, fiber intake, LDL, HDL cholesterol, fasting glucose; and 1.31 (CI: 1.09-1.58, $p<0.01$) in Model 3 (Model 2+CRP) (“Results”, page 8; Table 4, page 30).

These new analyses on both MDC-CC and DIRECT-METSIM cohorts showed that the relation between the incidence of diabetes and plasma follistatin did not change after adjustment with additional variables.

The new data are added in “**Abstract**” (page 4), “**Results**” under heading “PLASMA FOLLISTATIN LEVELS ASSOCIATE WITH THE RISK OF T2D” (page 7-8), **Table 2** (page 28), **Table 4** (page 30) and **Figure S1 legend** (page 58). The main analysis results are also highlighted here below in Table 2 and Table 4.

Table 2. Incidence of diabetes in relation to sex-specific quartiles of plasma follistatin levels in MDC-CC

	Q1	Q2	Q3	Q4	p for trend	HR per SD
Number of participants	1048	1049	1050	1048		
Incidence of diabetes n (n per 1000 p-y)	108 (5·16)	132 (6·54)	149 (7·52)	188 (9·87)		
Follistatin (NPX)[†]	17·27±1·21	23·92±1·07	30·06±1·08	42·22±1·23		
Model 1 HR	1	1·28 (0·99-1·66)	1·47** (1·15-1·89)	1·97*** (1·55-2·50)	<0·001	1·29*** (1·19-1·40)
Model 2 HR	1	1·12 (0·86-1·45)	1·21 (0·94-1·57)	1·45** (1·13-1·86)	0·003	1·31** (1·11-1·56)
Model 3 HR	1	1·10 (0·85-1·43)	1·16 (0·89-1·50)	1·35* (1·04-1·74)	0·020	1·24* (1·04-1·47)

Model 1: Adjusted for age and sex (n=4195). Model 2: Adjusted for age, sex, BMI, physical activity, alcohol intake, fiber intake, smoking habits, use of anti-hypertensive medications, systolic blood pressure, LDL, HDL cholesterol, use of lipid lowering medications, fasting glucose (n=4060). Model 3: Model 2 and CRP (n=4060). HR, hazard ratio. * $p < 0·05$, ** $p < 0·01$, *** $p < 0·001$. †Follistatin is expressed as linear Normalized Protein eXpression (NPX) AU for relative quantification according to Olink guidance.

Table 4. Incidence of diabetes in relation to quartiles of follistatin in the IMI-DIRECT-METSIM cohort

	Q1	Q2	Q3	Q4	p for trend	HR per SD
Number of participants	270	270	269	270		
Incidence of diabetes n (n per 1000 p-y)	10 (9.26)	13 (12.04)	11 (10.22)	19 (17.59)		
Follistatin (NPX)	2664.94±343.49	3447.24±205.82	4194.21±258.59	5725.14±1163.54		
Model 1 HR	1	1.69 (0.80-3.58)	1.49 (0.70-3.22)	2.61** (1.29-5.30)	0.011	1.35** (1.13-1.61)
Model 2 HR	1	1.12 (0.48-2.60)	1.19 (0.52-2.75)	2.25* (1.04-4.86)	0.029	1.28** (1.075-1.524)
Model 3 HR	1	1.13 (0.49-2.62)	1.23 (0.53-2.86)	2.34* (1.07-5.101)	0.0237	1.31** (1.09-1.58)

Model 1: Adjusted for age (n=1079).

Model 2: Adjusted for age, BMI, physical activity, alcohol intake, fiber intake, LDL, HDL cholesterol, fasting glucose (n=883).

Model 3: Model 2 and CRP (n=883).

HR, hazard ratio. * $p < 0.05$, ** $p < 0.01$

3. A major hypothesized pathway relates to the release of FFA from adipose tissue. However, it would be important to clearly demonstrate this relation in humans. While Suppl. Tables show correlations with FFA, these are univariate analyses, not adjusted for other factors.

Answer: We agree with the Reviewer that it is not only important to show the independent relationships of circulating follistatin with adipose tissue insulin sensitivity, but also with FFA levels. Please see the novel text on page 8, the last paragraph, which now reads:

“...plasma follistatin levels correlated positively with FFAs, measured before and during the OGTT and visceral fat mass. **Importantly, the relationships of plasma follistatin with fasting (std. $\beta=0.17$, $p=0.009$), 60 minutes (std. $\beta=0.26$, $p<0.0001$) and 120 minutes (std. $\beta=0.27$, $p<0.0001$) FFAs were independent of age, sex and total body fat mass.** Furthermore, plasma follistatin levels correlated negatively with adipose tissue insulin sensitivity ...”.

4. GWAS: the results of GWAS indicate the GCKR gene as a prominent locus. However, in contrast to the reporting, the strongest association is not observed for variants in this gene – rather the associations show the lowest p-value. Stronger associations have been observed for variants with lower MAF, e.g. ZNF333, ADAMTS14 or TMEM44. Although such variants are briefly discussed, it remains unclear why those genes were not followed further. It also remains unclear if the analysis in SUMMIT refers to another exploratory GWAS or a true replication of those variants identified in MDC-CC, the latter would be preferred.

Answer: We agree with the Reviewer that other genes identified may also contribute to the regulation machinery of plasma follistatin. The SUMMIT cohort is a true replication of the *GCKR* variants identified in MDC-CC, therefore we chose GCKR for our investigations. Furthermore, our unpublished RNA-seq data from liver cell lines show that follistatin expression is strongly affected by changes in GCK/GCKR. The effect of e.g. *ZNF333*, *ADAMTS14* or *TMEM44* on follistatin should be further investigated, however, it is outside the scope of the current study. We have now added this information in the “Discussion” (page 16, 2nd to last paragraph) which reads:

The effect of these identified genes strongly associated with circulating follistatin requires further investigation.

5. The authors miss to investigate the causal nature of the diabetes association using Mendelian randomization. This would be a logical step following the GWAS analysis. Importantly in this context, the identified GCKR gene is a major locus for triglyceride levels and also CRP. Thus potential pleiotropic effects versus mediators would need to be carefully considered.

Answer: There are several reasons why genetic variants may not be valid instrumental variables, such as pleiotropy (a variant affects risk factors on different causal pathways), linkage disequilibrium with a variant that influences another causal pathway, and population stratification.

Using a variant of *GCKR* in Mendelian randomization analysis is problematic because this variant of *GCKR* is the most pleiotropic variant in our exome sequencing study of close to 10,000 individuals (unpublished data in the DIRECT-METSIM study). On top of multiple metabolites reported in previous publications, we found several novel associations of *GCKR* variant with: 12 metabolites in the amino acid pathways, 2 in gamma-glutamyl amino acids, 2 in carboximide acids, 18 in glycerolipids, 50 in glycerophospholipids, 3 in sphingolipids, 5 in sphingomyelins, 1 hydroxy acid, 3 in keto acids, 2 in dicarboxylic acids, 2 in bile acids, 5 in cofactors and vitamins, 3 in xenobiotics, and 9 in other metabolites. Clearly a variant in *GCKR* is highly pleiotropic and affects several metabolic pathways that is against the major assumptions of the Mendelian randomization method. It is questionable if Mendelian randomization analysis could be applied using a variant in *GCKR*.

Minor points:

6. Abstract: The reporting of associations across different time spans of follow-up is irritating. It is unclear if this reflects a hypothesis of time varying strength of association or merely the fact that different cohorts were used.

Answer: The content of the Abstract is now revised following the Reviewer's suggestion (page 4, changes highlighted in red).

7. The c-statistics (page 7) indicate that follistatin does not improve diabetes prediction beyond established risk factors (model 2). This should be made clear from the wording in the results and also in the interpretation of the finding in the discussion (page 12).

Answer: We thank the Reviewer for the suggestion, and this information is now added in Results (page 7, last paragraph, the last sentence) and Discussion (page 12, 2nd paragraph, the last sentence).

8. Also, it remains unclear why not measures more suitable for survival data have been used in this context – a c-statistic from logistic regression doesn't account for varying time under risk of participants.

Answer: The C-statistics value is a measure of adequacy of fit for the binary outcomes. In this study, we presented both survival data and C-statistics.

9. It would be worthwhile to discuss findings for CRP in more detail. The correlation of follistatin with CRP seems quite substantial. While the authors adjust for CRP levels it would be informative to reflect if CRP needs to be seen as confounder or rather a mediator for a diabetes association. This seems also relevant in light of GCKR identified by GWAS, being also a major locus for CRP.

Answer: We thank the Reviewer for this suggestion, which is now addressed in new analysis using CRP as a variable for adjustment (Table 2 and 4). Addition of CRP for adjustment did not change the association analysis results.

Reviewer #3 (Remarks to the Author):

This is a very interesting analysis that aims to elucidate the relationship between follistatin and type 2 diabetes risk, as well as mechanisms underlying the associations of interest. The reviewer found this work interesting and of importance.

Specific comments:

1. The introduction left a mixed impression of the function of follistatin under physiological and pathophysiological conditions. For example, the authors suggested that follistatin may improve beta cell mass and counteract glucagon, whereas in other places the language suggested that this cytokine induces insulin resistance. Can the authors address this mixed functions of follistatin? Is it possible that, like leptin, follistatin may function as a marker of various physiological statuses?

Answer: We thank the Reviewer for this comment, and agree that the physiological role of follistatin and abnormally elevated follistatin and its effects under pathological conditions may represent different avenues. This is now discussed and presented in Discussion (page 12-13):

“Previous investigations have presented rather contradictory evidence on the effects of follistatin under physiological and pathophysiological conditions¹³⁻¹⁵. However, the physiological regulation of follistatin secretion and pathological effects of abnormally elevated follistatin may represent different avenues. Under normal physiological conditions, disruption of the GCKR-GCK complex, triggered by e.g. glucose and fructose²², stimulates follistatin secretion, which is regulated by insulin and glucagon. However, in an insulin resistant state, attenuated insulin signaling in the liver may lead to elevated follistatin secretion as previously shown in mouse models¹⁵. Abnormally elevated follistatin secretion may further exacerbate liver insulin resistance by promoting FFA production from adipose tissue and ultimately lead to NAFLD, possibly aggravating diabetes. The previous finding that follistatin increases beta cell proliferation during normal physiological conditions¹³ is perfectly in line with the need for increased insulin secretion to compensate for insulin resistance, and furthermore raises the intriguing possibility that follistatin plays a key role in mediating this signal from the liver to the pancreatic beta cell. Further studies are needed to understand

the role of follistatin in the cross talk between liver and pancreas in normal as well as pathophysiological conditions.”

2. Line 160-163. The association between follistatin and liver fat contents was not independent of visceral fat mass and leg fat mass. How could the authors suggest that the marker was independently associated with fatty liver in humans?

Answer: We thank the Reviewer for this comment and agree that we need to specify this point. For this please see page 9, paragraph 1, which now reads:

“The relationship of follistatin levels with liver fat content disappeared after further adjustment for visceral fat mass and leg fat mass, or for adipose tissue insulin sensitivity (Table S6). Thus, these findings suggest that circulating follistatin independently associates with adipose tissue insulin resistance in humans, and that the relationship of follistatin with fatty liver is possibly explained by effects of follistatin on insulin sensitivity of adipose tissue in the leg and the visceral compartment.”

3. Given that the authors also found significant genetic variants that predict the levels of follistatin levels, a logic next step is to perform Mendelian randomization to explore whether the epidemiological associations are likely causal or not in the MDC-CC cohort. This will make this work much more convincing.

Answer: We thank the Reviewer for this question, which was also raised by Reviewer 2. Please kindly refer to our answer to Question 4 from Reviewer 2 for details.

4. Were the CV estimates based on blind QC samples? Need to clearly indicate how the CVs were estimated.

Answer: CV was calculated per assay (i) using the assumption of a log-normal distribution. The CV was then averaged across panels.

$$CV_i = 100 * \text{sqrt} \left(e^{Sln_i^2} - 1 \right), \text{ where } Sln_i = \ln(2) * SD_i$$

This information is now added in the “**MATERIALS AND METHODS**” (page 18, first paragraph, with reference to www.olink.com for details).

As detailed on Olink, intra-assay variation (within-run) was calculated as the mean coefficient of variation (% CV) for 7 individual samples, within each of the 9 separate runs during the validation studies. Inter-assay variation (between-run) was calculated as the mean coefficient of variation (% CV), for the same 7 individual samples, between the 9 separate runs during the validation studies. Variation calculations were assessed on linearized values for 90 out of 92 analytes. Across 90 assays, the mean CV intra-assay

	Analytical measurement					Precision		
	pg/mL			log10		Intra-assay	Inter-assay	Inter-site
	LOD	LLOQ	ULOQ	Hook	Range			
Follistatin	977	977	1000000	2000000	3.0	10%	13%	14%

and inter-assay variations were observed to be 8% and 15%, respectively. The analytical measurement and precision on follistatin is shown in the table below.

Analytical Measurement; Limit of Detection (LOD), Lower Limit of Quantification (LLOQ), Upper Limit of Quantification (ULOQ), Hook, and Precision indicative of assay performance are shown for 92 analytes. Values below limit of detection were not reported (NR).

5. The results suggested that the proportional hazards assumption may not hold true. Have the authors examined this assumption in their analysis? It would be informative to plot HR by follow-up duration to understand the temporal pattern of this association.

Answer: This question was also addressed by Reviewer 2. Please kindly refer to our answer to Question 1 from Reviewer 2 for details.

6. Table 2. A few important variables were not considered in the multivariate adjustment, including physical activity, alcohol intake, BMI as an indicator of overall obesity, and diet quality.

Answer: This question was also addressed by Reviewer 2 in Question 2. Please kindly refer to the new analysis presented in response to Reviewer 2 for details.

7. Table 4 and Figure 2. The multivariate adjustment is minimal in these analyses.

Answer: We assume that the Reviewer refers to Table S4 and Figure 2. We absolutely agree with the referee that no adjustment (Table S4) and only a minimal adjustment (Figure 2) are being done here. Our intention is to first show the unadjusted data, followed by a very parsimoniously adjusted statistical model. We think that this approach allows the readers, step by step, to better understand the complex relationships of the parameters of interest, prior to showing the fully-adjusted multivariate models (e.g. Table S6).

REVIEWER COMMENTS

Reviewer #1 (Remarks to the Author):

The authors have been sufficiently responsive to my concerns. I would however recommend that the authors discuss the hypothesis that adipose tissue is the main target of follistatin in mice.

Reviewer #2 (Remarks to the Author):

Thank you for addressing all critique points.

Reviewer #3 (Remarks to the Author):

The review finds this work very interesting and believes that the work is important for the field. However, a few concerns/thoughts were raised during review. Most of them are quite minor.

1. It is proportional hazards regression. Don't forget the s.

2. We already have some cases where the then novel adipokines such as leptin were not causally associated with disease risk in humans even though they may contribute to disease etiology in animal studies. As such, the authors may want to conduct a Mendelian randomization analysis given that they know which SNPs predict the levels of follistatin. Such evidence will greatly strengthen the argument that this marker is worth further investigations if the MR analysis shows a positive link.

3. Need a statement regarding how the samples of cases and noncases were arranged in such a way that the biomarker results are not subject to any systematic bias.

4. Adjustments of fasting glucose and CRP are over-adjustments because they may operate in the pathways through which follistatin promotes diabetes risk.

5. Spearman correlations are much better than linear regressions to help readers understand the strength of associations intuitively because correlation coefficients are scale-free.

Overall, this is a very impressive study. And the reviewer wants to congratulate the authors for this achievement.

Elevated circulating follistatin associates with an increased risk of type 2 diabetes

Manuscript ID NCOMMS-20-42000

Response to Reviewer

REVIEWER COMMENTS

Reviewer #3 (Remarks to the Author):

The review finds this work very interesting and believes that the work is important for the field. However, a few concerns/thoughts were raised during review. Most of them are quite minor.

1. It is proportional hazards regression. Don't forget the s.

Answer: We thank the Reviewer for pointing it out, and have made the correction on page 4 (highlighted in red).

2. We already have some cases where the then novel adipokines such as leptin were not causally associated with disease risk in humans even though they may contribute to disease etiology in animal studies. As such, the authors may want to conduct a Mendelian randomization analysis given that they know which SNPs predict the levels of follistatin. Such evidence will greatly strengthen the argument that this marker is worth further investigations if the MR analysis shows a positive link.

Answer: We agree with the Reviewer that Mendelian randomization (MR) analysis would be helpful to further investigate causal relationships of follistatin with diabetes risk, by using the SNPs identified to be associated with follistatin levels in our study. However, using variants of *GCKR* in MR analysis, which we identified to be most strongly associated with follistatin levels, is very problematic, because these variants of *GCKR* were found to be

the most pleiotropic variants in our exome sequencing study of close to 10,000 individuals (unpublished data in the DIRECT-METSIM study). In this respect the variant rs780094 in *GCKR*, which most strongly associated with follistatin levels in our study, was found to associate with elevated fasting and random glycemia and increased risk of type 2 diabetes in several large studies. However, it also strongly associated with lower CRP, triglycerides (TG), LDL cholesterol levels and other diabetes risk parameters (<https://t2d.hugeamp.org/variant.html?variant=rs780094>). We have now also checked whether the same SNP associates with diabetes and metabolic traits in the MDC-CC study. We also found that it associates with CRP and TG levels, but not with incidence of diabetes. This is in contrast to the results from about 23 larger studies, where such a relationship with diabetes was observed. Thus, our MDC-CC study most probably is underpowered to perform MR analyses for these SNPs.

We did try to do a two sample MR analyses using SNPs associated with follistatin levels from the GWAS catalogue as exposure and data from the DIAMANTE GWAS meta-analysis of T2D (Mahajan et al, Nature Genetics, 2018) as outcome but found no significant association. However, the validity of the SNPs as instrumental variables is unclear given that only the *GCKR* locus was significant in our GWAS, as well as some apparent pleiotropy. We attach the results for the benefit of the Reviewer, but prefer not to add it to the manuscript, since no real conclusions can be made.

Outcome	Exposure	Method	n SNP	b	se	P value
T2D	Follistatin	MR Egger	6	0.05183856	0.55780583	0.93042538
T2D	Follistatin	Weighted median	6	-0.0102558	0.0605414	0.86548132
T2D	Follistatin	Inverse variance weighted	6	-0.1538512	0.1198639	0.19929979
T2D	Follistatin	Simple mode	6	-0.0131923	0.11990446	0.91667061
T2D	Follistatin	Weighted mode	6	0.03528085	0.04689369	0.48572518

We have now addressed the question about MR analyses in **DISCUSSION** (page 14-15, highlighted in red) which now reads:

Besides our previous and present findings on the impact of follistatin on the regulation of glucose and lipid metabolism in animals and *in vitro*, to further investigate causal relationships of follistatin with incidence of diabetes,

Mendelian randomization (MR) analysis would be an instrumental approach. Using variants of *GCKR* in MR analysis, which we identified to be most strongly associated with follistatin levels, is nevertheless very problematic, because these variants of *GCKR* were found to be the most pleiotropic variants in our exome sequencing study of close to 10,000 individuals (unpublished data in the DIRECT-METSIM study). In this respect the variant rs780094 in *GCKR*, which most strongly associated with follistatin levels in our study, was found to associate with elevated fasting and random glycemia and increased risk of T2D in several large studies. However, it also strongly associated with lower CRP, triglyceride, LDL cholesterol levels and other diabetes risk parameters (<https://t2d.hugeamp.org/variant.html?variant=rs780094>).

3. Need a statement regarding how the samples of cases and noncases were arranged in such a way that the biomarker results are not subject to any systematic bias.

Answer: The samples were consecutively aliquoted on plates, regardless of future diabetes status. The plates were analysed in random order. This information is now added in **MATERIALS AND METHODS**, page 18, the last paragraph (changes highlighted).

4. Adjustments of fasting glucose and CRP are over-adjustments because they may operate in the pathways through which follistatin promotes diabetes risk.

Answer: We agree with the Reviewer. Adjustment for glucose is certainly over-adjustment, and CRP is probably over-adjustment too. These covariates were added to the final model to make sure that factors in the subclinical phase of diabetes development were not responsible for the raised levels of follistatin. Or, in other words, that follistatin is a risk factor for development of diabetes, rather than vice versa, that development of diabetes in subclinical phase is a risk factor for increased follistatin. This information is now added in **DISCUSSION**, the second paragraph on page 12, which now reads:

We also included fasting glucose and CRP to the models to ensure that factors in the subclinical phase of diabetes development were not responsible for the raised levels of follistatin.

5. Spearman correlations are much better than linear regressions to help readers understand the strength of associations intuitively because correlation coefficients are scale-free.

Answer: Spearman is very useful in many situations. However, it has important limitations when other potential confounding factors need to be adjusted for – there is no established multivariate method that corresponds to Spearman correlations. The standardized betas, which are presented (for example on page 8), correspond to Pearson's correlation coefficients and can be interpreted in the same way as Spearman or Pearson correlations, with the difference that these estimates also are adjusted for confounding factors. Therefore, we would prefer reporting the standardized betas.

Overall, this is a very impressive study. And the reviewer wants to congratulate the authors for this achievement.

We sincerely thank the Reviewer for his/her kind appreciation and the very helpful comments that helped to improve the manuscript.

REVIEWER COMMENTS

Reviewer #3 (Remarks to the Author):

The reviewer thanks the authors for addressing the comments. The reviewer wants to point out that one can definitely calculate adjusted Spearman correlation coefficients, which are partial Spearman correlation coefficients. Spearman is better than linear regression beta coefficient or Pearson correlation coefficient when the distribution of variables is not normal.